# Once Upon an Input: Reasoning via Per-Instance Program Synthesis

**Adam Stein**[*]    **Neelay Velingker**[*]    **Mayur Naik**    **Eric Wong**
{steinad, neelay, mhnaik, exwong}@seas.upenn.edu
University of Pennsylvania

## Abstract

Large language models (LLMs) excel at zero-shot inference but continue to struggle with complex, multi-step reasoning. Recent methods that augment LLMs with intermediate reasoning steps such as Chain of Thought (CoT) and Program of Thought (PoT) improve performance but often produce undesirable solutions, especially in algorithmic domains. We introduce Per-Instance Program Synthesis (PIPS), a method that generates and refines programs at the instance-level using structural feedback without relying on task-specific guidance or explicit test cases. To further improve performance, PIPS incorporates a confidence metric that dynamically chooses between direct inference and program synthesis on a per-instance basis. Experiments across three frontier LLMs and 30 benchmarks including all tasks of Big Bench Extra Hard (BBEH), visual question answering tasks, relational reasoning tasks, and mathematical reasoning tasks show that PIPS improves the absolute harmonic mean accuracy by up to 8.6% and 9.4% compared to PoT and CoT respectively, and reduces undesirable program generations by 65.1% on the algorithmic tasks compared to PoT with Gemini-2.0-Flash. [2]

## 1 Introduction

Large-scale pretraining endows LLMs with the ability to recognize common concepts and perform many tasks in a zero-shot fashion but they still struggle with multi-step reasoning [1–3]. Recent advances in inference-time reasoning strategies such as Chain of Thought (CoT) and related work [4–6] have significantly improved LLMs' reasoning abilities. However, they remain unreliable [7–10] and unfaithful, meaning the final answer is correct for the wrong reasons [11–13].

Unlike LLM inference, program execution enforces precise, verifiable computation. Combining LLMs with program execution offers a promising reasoning method: the LLM handles perceptual inference, mapping raw input to structured form, while algorithmic reasoning is offloaded to an executable program [2, 14, 15]. Existing work on *neuro-symbolic learning* as well as methods such as Faithful Chain of Thought (FCoT) [16] and Program Aided Language Models (PAL) [17] adopts this approach, however, they use a single fixed program per task which causes problems when task instances are varied [18]. On the other hand, methods such as Program of Thought (PoT) [19] aim to enable LLMs to generate these programs in a zero-shot manner on an *instance-level*.

While flexible, these methods often produce undesirable programs, due to three challenges in instance-level program synthesis: (1) *open domain*: determining for a given instance if program synthesis is preferable to direct inference (via CoT) remains an open question, (2) *no task specifications*: there are no general specifications for how the correct program should behave to guide program search,

---

[*]These authors contributed equally to this work.

[2]Code for experiments and a demo is open-sourced at https://github.com/adaminsky/pips.

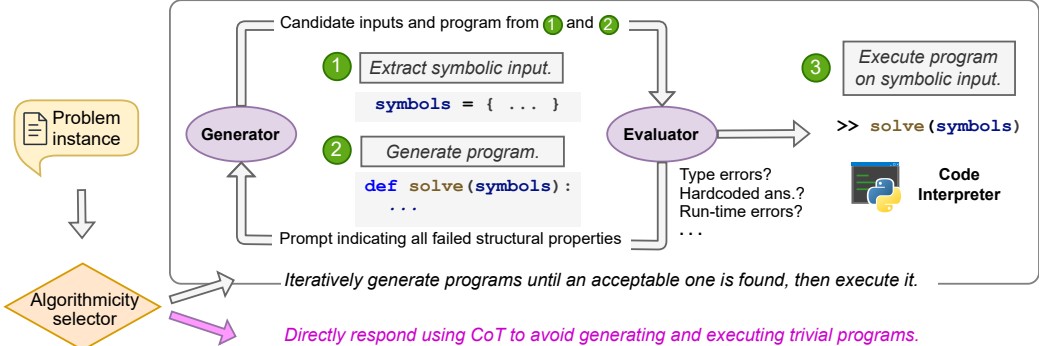

Figure 1: Overview of Per-Instance Program Synthesis (PIPS). PIPS addresses the open-domain nature of reasoning problems by selecting between synthesis and CoT at the instance-level, avoiding unnecessarily generating programs for non-algorithmic problems. For algorithmic problems, PIPS addresses the lack of task specifications by iteratively synthesizing programs using feedback based on *structural checks*. PIPS handles unstructured input via instance-specific symbolic extraction (step 1) before program synthesis (step 2). Figure 2 shows an example where an undesirable program is rejected before producing an acceptable one which gives the correct answer upon execution (step 3).

and (3) *unstructured input*: programs typically operate over structured input, but common reasoning problems are unstructured, requiring on-the-fly instance-level input understanding.

In this paper, we propose Per-Instance Program Synthesis, or PIPS, to solve reasoning problems at the instance-level. Figure 1 illustrates how PIPS addresses the aforementioned challenges. To address the open domain nature of such problems, PIPS introduces an instance-level confidence metric to decide if the LLM is better suited to solving the instance with program synthesis or direct inference. It then iteratively generates and evaluates programs using feedback based on structural checks (e.g. non-triviality, syntax, type errors), specifically designed to avoid the collapse to trivial solutions and ensure well-formed computation. To handle unstructured input, PIPS explicitly performs instance-specific symbolic extraction before synthesis, decoupling program search from perceptual inference. Unlike other iterative refinement and debugging methods for code generation [15, 20, 21], PIPS does not require explicit test cases, examples, or other forms of task specifications.

Our experiments demonstrate that PIPS significantly reduces the amount of undesirable programs produced compared to baselines, and this results in improved accuracy. Notably, PIPS reduces undesirable programs by 65.1% for algorithmic benchmarks in the Big Bench Extra Hard (BBEH) suite [7] as well as 7 additional tasks including visual question answering and mathematical reasoning, resulting in an 8.6% absolute improvement in harmonic mean accuracy over PoT. We also show that our confidence metric allows us to correctly switch between CoT and program synthesis for 65% of cases, resulting in PIPS matching CoT accuracy for majority non-algorithmic tasks and even improving performance on majority algorithmic tasks.

Our contributions are as follows:

- Per-Instance Program Synthesis (PIPS): An iterative program synthesis method guided by instance-specific feedback on program structure properties, improving reasoning by addressing the challenges of per-instance program synthesis methods.
- Synthesis Confidence Metric: We study the tradeoff between program synthesis and CoT for answering reasoning problems, and we propose a synthesis confidence metric which predicts, prior to generation, which approach is more likely to yield a correct solution for a given model.
- State-of-the-art accuracy: PIPS improves code utilization by 65.1% on algorithmic questions leading to an 8.6% improvement in harmonic mean accuracy over PoT across 30 frontier tasks.

## 2 The Challenges of Per-Instance Program Synthesis

A reasoning problem is a pair $(x, y) \in \mathcal{X} \times \mathcal{Y}$ where $x$ is a raw query consisting of unstructured data such as natural language text or an image, and $y$ is the answer which we assume is represented

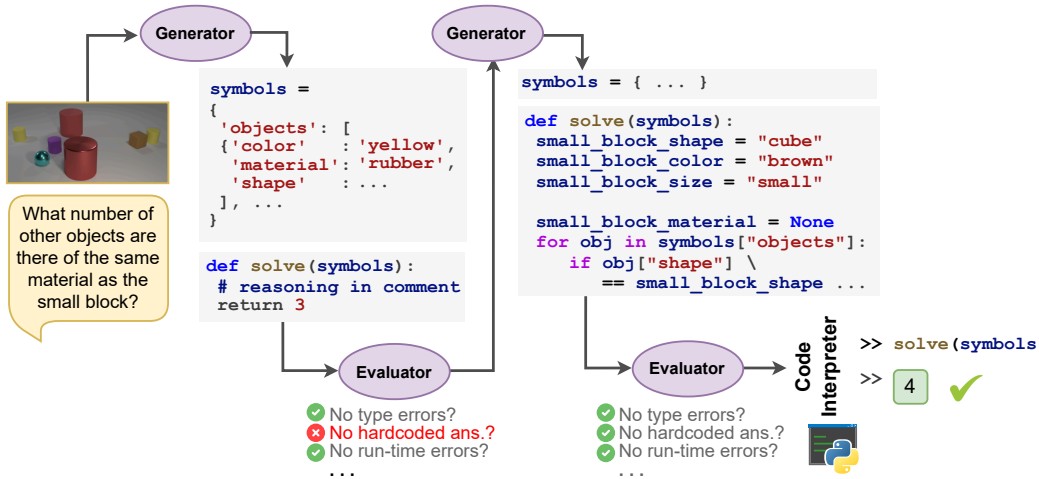

Figure 2: Example illustrating two iterations of the synthesis loop in PIPS.

symbolically (something that could be output by a program). Existing methods for generating instance-level programs for solving reasoning problems, such as PoT, produce a program $P : \varnothing \to \mathcal{Y}$ to compute the correct answer. These programs take no input, similar to a main function, but can be highly general since they are often generated in a Turing-complete language such as Python.

This problem-solving strategy naturally introduces three challenges. In short, algorithmic problems are well-suited to program synthesis but non-algorithmic problems are inherently ill-suited; producing a program cannot be guided by traditional specifications; and interfacing the unstructured input from the problem with the program requires on-the-fly instance-level perceptual understanding.

## 2.1 Non-Algorithmic Problems

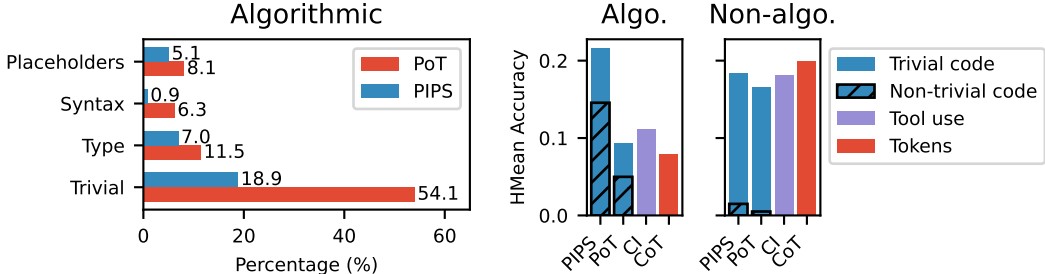

(a) Fraction of output code with issues: *trivial* (hard-coded answer), *type* (wrong return type), *syntax* (parser error), and *placeholders* (incomplete code).

(b) Per-instance synthesis baseline performance on mostly algorithmic vs. non-algorithmic problems. Shading indicates fraction of correct solutions with non-trivial code.

Figure 3: Failures in existing approaches to per-instance program synthesis with Gemini-2.0-Flash.

The per-instance synthesis setting poses a tradeoff between program synthesis and direct inference reliability. In some cases, it is more reliable for an LLM to solve a problem through token-based reasoning (like CoT) than through program synthesis. Examples include tasks like emotion understanding and summarization, which are not traditionally considered "algorithmic". We find that when existing per-instance synthesis methods are applied to these non-algorithmic settings, they will often output trivial programs that unnecessarily invoke the Python interpreter.

To study the *algorithmicity* of the problem instances across a variety of datasets, we design an LLM-based classifier to determine if an instance is algorithmic, meaning it could be solved by executing a non-trivial program. Using this classifier, we split our suite of datasets (which includes all 23 tasks of BBEH as well as 7 additional tasks) into *algorithmic* ones which have a majority of algorithmic

Figure 4: Two programs generated with PoT illustrating program synthesis failures. Part (a) shows a trivial program where two variables are initialized to zero, but then several steps of reasoning are performed in comments, leading to their values being hard-coded rather than computed with code. Part (b) shows an input-free program to process the input image itself which would be better done using the LLM's perceptual inference. Both programs result in the wrong answer. The corresponding programs produced by PIPS yield the correct answer in both cases and are shown in Appendix B.

instances, and *non-algorithmic* ones which have a majority of non-algorithmic instances. Details of our classifier and the full task split is provided in Appendix A. Overall, we find that not all tasks are purely algorithmic or non-algorithmic, meaning that algorithmicity is best determined on the instance-level. In addition, as Figure 3b shows, code execution via PoT prompting significantly improves performance on the algorithmic tasks, while having a slightly negative impact for the non-algorithmic tasks. We follow the recommendation from Kazemi et al. [7] to evaluate harmonic mean accuracy since it is a challenging metric that requires improvements on the hardest tasks to improve aggregate performance. For these non-algorithmic problems, PoT rarely outputs well-formed code. Instead of producing superficial code for non-algorithmic problems, we could skip program synthesis entirely with no harm to performance or interpretability.

## 2.2  Program Search without Specifications

Existing instance-level code generation methods often use the first generated program [19], or use minimal forms of program search due to the lack of any explicit specifications for how the correct program should behave. We find that this frequently leads to "trivial" programs which have an explicit return value hard-coded. Figure 3a shows that over 50% of PoT's outputs on algorithmic tasks fall into this category. The lack of behavioral specifications leaves the LLM the option of solving the task through direct inference with the final answer wrapped in a program. Figure 4a shows a trivial program produced by PoT. Additionally, 6.3% of programs have syntax errors and 11.5% return the wrong type which is why it is undesirable to always settle with the first generated program. Per-instance synthesis approaches suffer from these problems since they lack the necessary task specifications to perform traditional program search. Our full evaluation criteria including well-formed programs, type errors, and syntax errors is in Appendix G.

## 2.3  Interfacing Programs with Unstructured Data

The use of programs is best when their input is a well-defined symbolic structure as opposed to raw data (e.g. a paragraph of text or an image). Since existing methods generate input-free programs, they must either hard-code the necessary structured input into the program (using the LLM's perceptual understanding) or use the code to process the unstructured input. Figure 4b shows an example of a

**Algorithm 1** PIPS: Synthesis Loop

---

**Require:** Input $x \in \mathcal{X}$, symbolic extractor $c$, maximum iterations $k$
**Ensure:** Program $P^*$ such that $P^*(c(x)) \approx y$
 1: Initialize $i \leftarrow 0$
 2: Extract symbols: $r_0 \leftarrow c(x)$
 3: Generate initial program: $P_0 \leftarrow \texttt{LLM}(x, r_0)$
 4: **for** $i = 1$ to $k$ **do**
 5:     Evaluate program: $F_i \leftarrow E(P_i, r_i, x)$
 6:     **if** $F_i = \texttt{Pass}$ **then**
 7:         **return** $P_i$
 8:     **end if**
 9:     Update symbols if they have associated errors: $r_{i+1} \leftarrow c(x; \{r_j\}_0^i, \{P_j\}_0^i, \{F_j\}_0^i)$
10:     Generate revised program: $P_{i+1} \leftarrow \texttt{LLM}(x; \{r_j\}_0^{i+1}, \{P_j\}_0^i, \{F_j\}_0^i)$
11: **end for**
12: **return** $P_k$ {Fallback if none pass}

---

program produced by PoT on the CLEVR task [22] wherein the program attempts to parse objects from an image using code instead of leveraging the strong perceptual understanding abilities of the LLM itself. Executing the program leads to an error caused by referencing a non-existent file.

We find that 12.7% of well-formed PoT code solutions to the CLEVR and Leaf multimodal datasets use the OpenCV or Pillow libraries, representing a fundamentally brittle approach to input understanding.

# 3 Per-Instance Program Synthesis (PIPS)

This section addresses the above three challenges with an approach to synthesize programs on a per-instance basis without task specifications. We use the general problem-solving structure of $y = P(c(x))$ where $c : \mathcal{X} \to \mathcal{R}$ is a function mapping from raw input space to permissible program inputs and $P : \mathcal{R} \to \mathcal{Y}$ is a program in a Turing-complete programming language. The next sections describe how we address the previous challenges with this problem-solving framework.

## 3.1 Selective Program Synthesis

While program synthesis allows for more faithful and sophisticated reasoning, there may still be specific problem instances where CoT should be used. The decision between synthesis and CoT for solving an instance depends on both the algorithmic nature of the problem and the model's capabilities in the two respective solving approaches. Formally, given a reasoning problem $(x, y) \in \mathcal{X} \times \mathcal{Y}$, we must choose between two strategies: (1) direct CoT reasoning via $\hat{y} = M_{\text{cot}}(x)$, where $M_{\text{cot}}$ denotes chain of thought inference, or (2) program synthesis via $\hat{y} = P(c(x))$, where $P$ is a synthesized program and $c(x)$ is a symbolic abstraction of the input $x$.

Since the decision depends on the model's own problem-solving abilities, we elicit it from the model before it begins reasoning. Our self-prompting method uses ten criteria to choose between CoT and program synthesis. Each criterion results in a confidence score from the model, forming a vector $S(x) = (p_1(x), \ldots, p_{10}(x)) \in [0, 1]^{10}$. These criteria assess factors like ease of formalization, expected execution success, and robustness of logic. For reasoning models, we include ten additional criteria relating to their reasoning abilities. Motivated by prior work demonstrating that LLMs can accurately estimate the likelihood that their answer to a question is correct [23], we hypothesize that these criteria, which are agnostic to the downstream task, can be strong predictors for the LLMs success in per-instance synthesis. Full details of the criteria are in Appendix H.2.

Given the model's own assessment of its abilities through these probing questions, the final switch decision can either be derived in a fully zero-shot manner, or a held-out calibration set can be used to derive the decision from a learned logistic classifier, enabling the switch to leverage problem solving experience for higher accuracy.

## 3.2 Program Synthesis without Task Specifications

Traditional program synthesis searches the space of programs guided by a task specification (either input-output examples or a logical specification) which determine which programs are acceptable and which are not. To perform program synthesis with just a single input, we leverage auxiliary forms of specifications based on generally undesirable failure modes of code generation. Formally, given an input $x \in \mathcal{X}$ which may consist of a request such as "How is Bob related to Alice" as well as image inputs like an image of a family tree, we want to find a program $P$ such that $P(c(x)) = y$ where $y$ is the true answer. Our goal is to search for a $P$ to optimize the following:

$$\min_P M(P \mid x, c(x)) + \lambda E(P, c(x); x)$$

where $M$ is an LLM and $E$ is a program evaluator which checks various properties of code based on static and dynamic analysis. To design the program evaluator $E$, we study the common failure modes of LLM code generation in this setup without a feedback loop.

PIPS is an approach to solve for $P$, described in Algorithm 1 and illustrated in Figure 1. We design $E$, consisting of an LLM and a program interpreter, to flag any program $P_i$ matching any of the aforementioned patterns. If an issue is detected, $E$ produces structural feedback to aid in fixing the problem. Issues with the structured input $r_i$ result in a revised program input $r_{i+1}$ conditioned on the history of programs and inputs. If any of the evaluator feedback pertains to the synthesized program, then the generator produces a revised program $P_{i+1}$ conditioned on the feedback past programs. This iterative refinement continues for at most $k$ steps, or until $E(P_i, r_i, x)$ detects no issues. We note that at the initial step $k = 0$, $M$ is prompted to produce a program that does not contain these patterns.

### 3.3 Converting Raw Data to Symbolic Program Input

To address the issue of interfacing discrete programs with unstructured data, we explicitly abstract the unstructured data as structured program input before program generation. The input abstraction is done with the mapping $c$, and we use an LLM for this task due to their ability for understanding general unstructured data. Concretely, the LLM processes the input to identify salient entities, their attributes, and the relationships between them that are pertinent to solving the instance. This requires the LLM to infer an ad hoc schema for the JSON structure, tailored to the specific semantics of the input, rather than have this schema be predetermined. While allowing for an LLM to decide how to abstract the input into a structured form is highly general, this can lead to mistakes or missed information from the input in this step. The synthesis loop described above iteratively fixes such issues in the data abstraction in coordination with program generation.

## 4 Experiments

This section studies whether PIPS addresses the challenges of per-instance program synthesis. **RQ1** studies the empirical performance of PIPS, and **RQ2**, **RQ3**, and **RQ4** correspond to each respective challenge laid out in Section 3.

### 4.1 Setup

**Datasets** We evaluate our approach using 23 tasks sourced from the Big Bench Extra Hard (BBEH) benchmark [7]. These tasks span topics such as geometric understanding, deductive logical reasoning, and commonsense understanding. Furthermore, we extend this study to the visual reasoning tasks CLEVR [22] and Leaf [24], the relational reasoning task CLUTTR [25], and four mathematical reasoning tasks of OmniMath [26]. For all datasets, we reserve a random sample of 20% of the data for calibration of our confidence switch, and we evaluate on the remaining 80% of the data. We also evaluate the generalizability of a trained confidence switch in a leave-one-dataset out evaluation scheme as well as show that a fully zero-shot confidence switch is also highly effective in Appendix H.1.

**Models** To evaluate our setup across a variety of different frontier LLMs, we use Gemini-2.0-Flash [27], GPT-4.1-mini [28], and o4-mini [29]. Gemini-2.0-Flash is a multimodal LLM, GPT-4.1-mini is a lightweight general-purpose LLM, and o4-mini is a multimodal LLM that was trained for "reasoning" with a long CoT before its final response.

**Baselines** We evaluate PIPS against Program of Thought (PoT) [4], Chain of Thought [4], and a code interpreter tool-use agent. PoT involves prompting an LLM to generate Python code to solve the

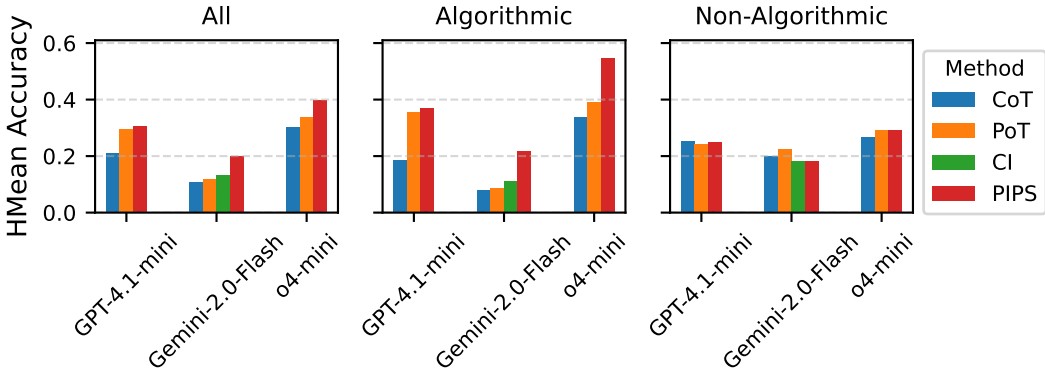

Figure 5: Harmonic mean accuracy over all 30 datasets (left), on the 17 majority algorithmic tasks (middle), and on the 10 majority non-algorithmic (right) for PIPS and baselines using three state-of-the-art models. The breakdown per task per model is shown in Table C.3, Table C.2, and Table C.4.

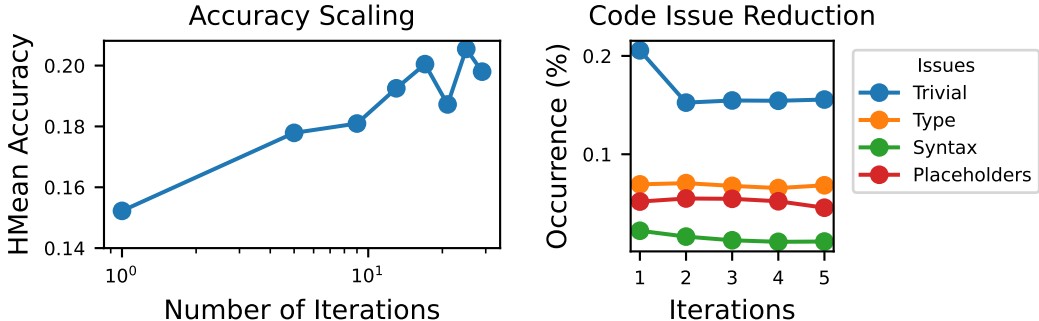

(a) Scaling of harmonic mean accuracy across all datasets with more synthesis iterations.

(b) Decrease in code issues with more synthesis iterations.

Figure 6: Accuracy and code quality scaling with more iterations of PIPS with Gemini-2.0-Flash.

problem and then executing the code to get a final answer. Gemini Code Interpreter (CI) [27] is an API tool that allows the model to synthesize and execute code in an *agentic* manner before producing its final response.

We evaluate another agentic baseline called PoT-retries which performs PoT, but regenerates the code until the code executes without any errors, as well as CodeAct [30], and Buffer-of-Thoughts [31] for Gemini-2.0-Flash in Appendix D.

### 4.2 RQ1: Does PIPS outperform baselines across datasets?

We compare PIPS to baselines in terms of overall performance. Harmonic mean aggregated results of PIPS compared to baselines over all 30 datasets as well on just the algorithmic and non-algorithmic tasks are shown in Figure 5. Overall, we see an absolute improvement of 8.6% in harmonic mean accuracy over PoT, with up to a 23.7% improvement in absolute accuracy over PoT (on BBEH Boolean Expressions) for Gemini-2.0-Flash, a 0.8% absolute improvement over PoT for GPT-4.1-mini, and a 5.7% absolute improvement over PoT for o4-mini. From the middle plot, we can see that PIPS provides significant improvements over the baselines (up to 15.9% in absolute harmonic accuracy for o4-mini) on the majority algorithmic problems, while not degrading in accuracy for the non-algorithmic tasks. Full results are included in Appendix C.

We further study the performance of PIPS as the number of feedback iterations $k$. Even at $k = 0$, meaning the evaluator is not used, PIPS outperforms PoT by a harmonic mean difference of 5.6% and CI by 3.7%. This gap widens as we scale $k$, as shown in Figure 6a. Iteration successfully leads to more well-formed programs which subsequently improves correctness.

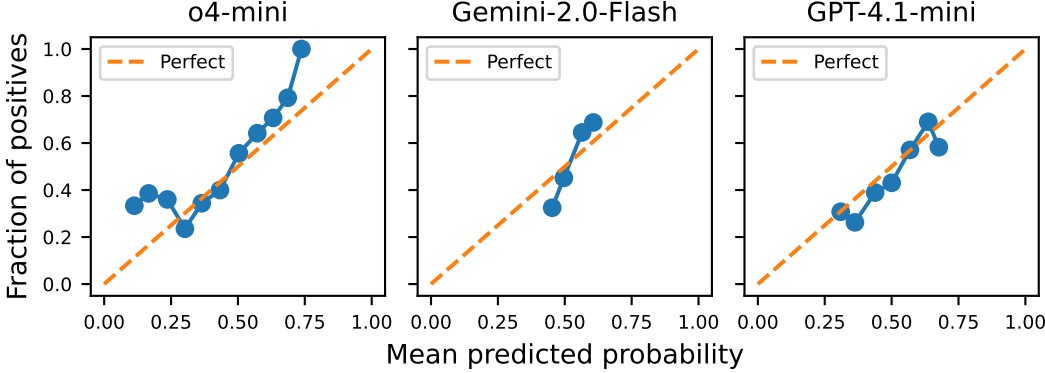

Figure 7: Calibration curve of our selection method between program synthesis and CoT. We only consider questions where the choice between synthesis and CoT determines answer correctness, and a positive instance is one correctly solved by code. Therefore, a score of 0.8 should mean an 80% chance of solving correctly with code and a 20% chance of solving correctly with CoT.

We report the average cost of PIPS and baselines in Appendix E and show that PIPS additionally achieves lower cost than other iterative approaches.

Table 1: Ablation study for BBEH tasks.

| Method | HMean Accuracy (%) |
| --- | --- |
| PIPS | 20.8 |
| PIPS (no switch) | 18.3 |
| PIPS-0 (no switch) | 12.9 |
| PIPS-0 (no switch, no symbols) | 4.3 |

### 4.3 RQ2: How effective is switching between synthesis and CoT on the instance-level?

In this RQ, we investigate whether our switch can effectively decide between synthesis and CoT before committing to either option. As described in Section 2.1, this is important since non-algorithmic problems almost always result in trivial programs which are equivalent to CoT, but incur an unnecessary call to the Python interpreter. Note that non-algorithmic instances occur even in majority algorithmic tasks. We show in Appendix F that encouraging non-trivial code for non-algorithmic instances with our iterative search process can reduce performance.

To evaluate our switch, we focus on cases where it affects the outcome—i.e., when either PIPS or CoT is correct, but not both. These comprise 24.8% of all samples across 30 benchmarks. For Gemini-2.0-Flash, the switch selects the correct method 65.3% of the time, yielding a 2.2% absolute gain in harmonic mean accuracy (Table 1). Example switch decisions are shown in Appendix H.3.

Furthermore, we investigate the level of calibration of our switching method. As illustrated in Figure 7, the switch is indeed well calibrated. Notable sources of deviation occur at the extremes. We further study the marginal contributions of each $p_i$ with respect to $S$ in Appendix H.4. Importantly, these results demonstrate the usefulness of an LLM's intrinsic understanding of its own problem solving abilities for choosing when program synthesis is appropriate on an instance-level.

### 4.4 RQ3: Does PIPS improve code quality and correctness?

In this RQ, we seek to study how PIPS improves code quality. First, we analyze the performance results to ensure that PIPS produces programs that are meaningful. To verify alignment, we filter PIPS and all baselines' code outputs by those that are well-formed according to our evaluator criteria described in Section 3.2. As seen previously in Figure 3b, PIPS produces significantly more well-formed programs than PoT when solving algorithmic problems, and the discrepancy in absolute percentage points can go up to 53.7% as seen on the Temporal Sequence task. See Appendix C for

Table 2: Performance boost from PIPS fixing PoT's code issues on BBEH algorithmic tasks. Boost reflects accuracy gain on samples where PIPS corrected PoT errors.

| Issue Fixed by PIPS | Performance Boost ($\Delta$ Acc %) | Samples Fixed |
|---|---|---|
| Syntax Errors | 20.0 | 200 |
| Wrong Return Type | 16.8 | 297 |
| Placeholders | 7.2 | 194 |
| Hardcoded Answers | 5.2 | 1138 |

the percent of well-formed programs produced by PIPS and PoT for each of the 30 datasets. Table 2 studies the marginal impact of each type of fix on BBEH tasks.

Next, we focus mainly on questions that are considered algorithmic, as decided by the classifier described in Section 2.1. Focusing only on algorithmic samples across all benchmarks, Figure 3a demonstrates that PIPS can significantly reduce the issues exhibited in PoT. For example, the number of trivial programs is reduced by as much as 75.6%. Type and syntax issues are reduced by 49.2% and 86.8%, respectively. Lastly, the number of programs with placeholders are reduced by 36.3%. We study this more closely as iterations scale in Figure 6b, where we show as $k$ increases from 1 to 5, the percent occurrence of these undesirable properties decreases.

### 4.5 RQ4: Does PIPS reduce ineffective handling of structured data?

Unlike existing per-instance code generation methods, PIPS produces programs that take an explicit structured input. For instance, for image input, PIPS may extract relevant objects from the image and pass these objects to the program as a list. To determine if this explicit separation of the data and logic of a program reduces issues relating the incorrect handling of structured data, like that shown in Figure 4b, we perform a comparison of the code produced by PIPS with PoT. While 12.7% of PoT well-formed code to the multimodal benchmarks (CLEVR and Leaf) used the OpenCV or Pillow libraries which are for image processing, our method never tries to manually process images.

We also perform an ablation as shown in the bottom two rows of Table 1, where we see that the use of explicit function inputs in PIPS leads to a 4% harmonic mean improvement on BBEH. Without explicit inputs, PIPS-0 (no symbols), produces a single input-free function, but first performing structured input extraction before code generation has a significant performance improvement.

## 5 Related Work

**Reasoning with Code Generation.** LLMs have been used to generate structured symbolic representations—such as semantic parses [14, 16, 32] or domain-specific programs in PDDL, SMT, or Datalog—to enable external reasoning. These approaches rely on hand-crafted prompts and fixed DSLs that limit generality and expressiveness. Others prompt LLMs to produce executable code in general-purpose languages to solve problems directly [17, 19, 33], enabling stronger abstraction and reuse. However, such methods typically rely on few-shot prompts or example-based verification (as in Programming-by-Example) [15, 34, 35], limiting their applicability to tasks with clear specs or test cases. In contrast, PIPS performs instance-level program synthesis without requiring DSLs, specs, or handcrafted templates, and uses structural feedback to iteratively refine programs.

Approaches which prompt an LLM to produce code to solve a problem have been used in several domains beyond math and text-based reasoning questions. ViperGPT and followup work tackle visual question answering problems [36, 37], Voyager applies to game playing [38], and Code as Policies focuses on the application of robot control [39]. Recently, general systems such as CodeAct [30] and OpenCodeInterpreter [40] have been proposed for solving problems via code generation similar to the previously mentioned solutions for each task.

**Test-Time Optimization for Reasoning.** Prompting strategies like Chain of Thought [4, 41] and Tree of Thought [42] enhance LLM reasoning by decomposing problems or exploring multiple inference paths. Methods such as Hypothesis Search [15] and Self-Discover [43] further improve performance by searching over program hypotheses or reasoning formats. Recent work also explores LLMs as code evaluators [44, 45], but often requires human supervision or task-specific tuning. PIPS differs

by selecting between direct inference and code execution using a learned confidence signal, achieving robust and adaptable test-time reasoning with minimal assumptions.

# 6 Limitations and Conclusion

In this paper, we focus on simple structural code properties since they occur often in generated code. Further work is needed to determine if there are more undesirable patterns in LLM-generated code. In addition, PIPS does not optimally handle problems which are best solved partly with CoT and partly with program synthesis. Future work can tackle methods for problem decomposition and composing program synthesis with other forms of reasoning. Finally, while PIPS offers interpretable reasoning when using code, the conversion of the input to symbolic form still lacks faithfulness guarantees.

We introduced Per-Instance Program Synthesis (PIPS), a method that dynamically synthesizes reasoning programs by leveraging general structural feedback. By focusing synthesis on the instance-level, rather than the task-level, PIPS significantly outperforms prior code-based reasoning approaches such as PoT as well as purely textual reasoning via CoT, and produces much less *trivial* code than prior work. The efficacy of PIPS for solving the most challenging reasoning problems through program synthesis underscores the promise of synthesis as a powerful means of enabling complex reasoning, in addition to the current paradigm of CoT-driven reasoning.

## Acknowledgments and Disclosure of Funding

We thank Mayank Keoliya for his feedback and help with additional baselines during the rebuttal period.

This research was supported by the ARPA-H program on Safe and Explainable AI under the award D24AC00253-00, an NSF Graduate Research Fellowship, a Google Research Award, and a gift from AWS AI to ASSET (Penn Engineering Center on Trustworthy AI).

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

# A  Algorithmic Split of Datasets

We show the split of the BBEH datasets into algorithmic and non-algorithmic datasets in Table A.1. This split of datasets into the two groups is used in analyzing the results in the main body of the paper.

Table A.1: Percent of algorithmic problems in each dataset. We call the datasets with a majority of algorithmic problems the *algorithmic* datasets and the other datasets the *non-algorithmic* datasets.

| Dataset | % Algorithmic |
|---|---|
| **Non-algorithmic Datasets** | |
| Leaf | 0.000 |
| Disambiguation qa | 0.000 |
| Sarc triples | 0.000 |
| Nycc | 0.000 |
| Movie recommendation | 0.000 |
| Hyperbaton | 0.015 |
| Geometric shapes | 0.040 |
| Causal understanding | 0.040 |
| CLEVR | 0.155 |
| Linguini | 0.175 |
| Omnimath-4 | 0.305 |
| Sportqa | 0.345 |
| Omnimath-3 | 0.410 |
| **Algorithmic Datasets** | |
| Clutrr | 0.750 |
| Spatial reasoning | 0.810 |
| Omnimath-2 | 0.835 |
| Buggy tables | 0.940 |
| Web of lies | 0.940 |
| Boardgame qa | 0.955 |
| Object properties | 0.960 |
| Boolean expressions | 0.960 |
| Time arithmetic | 0.970 |
| Word sorting | 0.980 |
| Dyck languages | 0.980 |
| Temporal sequence | 0.990 |
| Omnimath-1 | 0.995 |
| Object counting | 0.995 |
| Zebra puzzles | 0.995 |
| Multistep arithmetic | 1.000 |
| Shuffled objects | 1.000 |

# B  PIPS Programs for Figure 4

The two corresponding solutions with PIPS to the problems shown in Figure 4a and Figure 4b are included in Figure B.1 and Figure B.2 respectively.

# C  Full Results

The full results over all 30 datasets for Gemini-2.0-Flash, GPT-4.1-mini, and o4-mini and included in Table C.3, Table C.4, and Table C.2 respectively. Results for an open-weights model, Qwen3-235B-A22B, for BBEH tasks is included in Table C.5.

```
1   symbols = {'objects': [{'color': 'green', 'material': 'rubber', 'shape': 'cube',
    ↪    'size': 'large', 'x': 0.1, 'y': 0.4},
2       {'color': 'blue', 'material': 'metal', 'shape': 'sphere', 'size': 'large',
        ↪    'x': 0.4, 'y': 0.3},
3       {'color': 'yellow', 'material': 'metal', 'shape': 'cube', 'size': 'large',
        ↪    'x': 0.6, 'y': 0.3},
4       {'color': 'cyan', 'material': 'rubber', 'shape': 'cylinder', 'size': 'small',
        ↪    'x': 0.7, 'y': 0.3},
5       {'color': 'brown', 'material': 'rubber', 'shape': 'sphere', 'size': 'large',
        ↪    'x': 0.5, 'y': 0.6}]}
6
7   def solve(symbols):
8       """
9       Calculates the number of large objects in front of the cyan object and behind
        ↪    the brown matte ball.
10
11      Args:
12          symbols: A dictionary containing information about the objects in the image.
13
14      Returns:
15          The number of large objects in front of the cyan object and behind the brown
            ↪    matte ball.
16      """
17
18      cyan_x = next((obj["x"] for obj in symbols["objects"] if obj["color"] ==
        ↪    "cyan"), None)
19      brown_x = next((obj["x"] for obj in symbols["objects"] if obj["color"] ==
        ↪    "brown"), None)
20
21      count = 0
22      for obj in symbols["objects"]:
23          if obj["size"] == "large":
24              if obj["x"] < cyan_x and obj["x"] > brown_x:
25                  count += 1
26
27      return count
```

Figure B.1: Final symbols and generated program from PIPS for the instance shown in Figure 4a. Executing the code from PIPS results in the correct answer of 1.

# D   Additional Baselines

In this section, we compare PIPS with an iterative refinement version of PoT, CodeAct [30], and Buffer-of-Thoughts (BoT) [31]. The method PoT-retries refers to our modified version of PoT which regenerates the program if the produced program results in an execution error. We use the CodeAct implementation provided with the smolagents library from huggingface. Finally, we use the code for BoT from Yang et al. [31] for this baseline and we additionally fixed several bugs which previously resulted in a high failure rate.

Results for PoT-retries, CodeAct, BoT, and PIPS are included in Table D.9. We show results using Gemini-2.0-Flash over all BBEH tasks since BoT does not support the multimodal tasks.

# E   Reasoning Costs

In addition to comparing the costs of different methods for using code for solving challenging reasoning problems, we also compare the cost in terms of number of tokens and dollar cost. Table E.10 shows average input, output, and dollar cost averaged over all 30 datasets for Gemini-2.0-flash. We find that the iterative approaches (CodeAct and Buffer of Thoughts) increase cost by more than 10X

```
1   symbols = {'objects': [
2       {'color': 'purple', 'material': 'rubber', 'shape': 'sphere', 'size': 'large',
        ↪   'x': 0.0},
3       {'color': 'purple', 'material': 'metal', 'shape': 'sphere', 'size': 'large',
        ↪   'x': 1.0},
4       {'color': 'blue', 'material': 'rubber', 'shape': 'cube', 'size': 'small', 'x':
        ↪   2.0},
5       {'color': 'gray', 'material': 'rubber', 'shape': 'cylinder', 'size': 'small',
        ↪   'x': 2.5},
6       {'color': 'brown', 'material': 'rubber', 'shape': 'cube', 'size': 'large',
        ↪   'x': 4.0},
7       {'color': 'yellow', 'material': 'rubber', 'shape': 'sphere', 'size': 'large',
        ↪   'x': 1.5},
8       {'color': 'brown', 'material': 'metal', 'shape': 'sphere', 'size': 'small',
        ↪   'x': 3.5},
9       {'color': 'red', 'material': 'metal', 'shape': 'cube', 'size': 'small', 'x':
        ↪   2.0},
10      {'color': 'red', 'material': 'metal', 'shape': 'cube', 'size': 'large', 'x':
        ↪   5.0}]}
11
12  def solve(symbols):
13      """
14      Finds the color of the tiny shiny object behind the big ball to the right of
        ↪   the big metallic thing behind the big brown cube.
15
16      Args:
17          symbols (dict): A dictionary containing information about the objects in
            ↪   the image.
18
19      Returns:
20          str: The color of the object.
21      """
22
23      objects = symbols["objects"]
24
25      # 1. Find the big brown cube
26      big_brown_cube = next((obj for obj in objects if obj["shape"] == "cube" and
        ↪   obj["color"] == "brown" and obj["size"] == "large"), None)
27
28      # 2. Find the big metallic thing (red cube) to the right of the big brown cube
29      big_metallic_thing = next((obj for obj in objects if obj["shape"] == "cube"
        ↪   and obj["color"] == "red" and obj["size"] == "large" and obj["x"] >
        ↪   big_brown_cube["x"]), None)
30
31      # 3. Find the closest big ball
32      closest_big_ball = min((obj for obj in objects if obj["shape"] == "sphere" and
        ↪   obj["size"] == "large"), key=lambda obj: abs(obj["x"] -
        ↪   big_metallic_thing["x"]))
33
34      # 4. Find the tiny shiny object (gold sphere) behind the big ball
35      tiny_shiny_object = next((obj for obj in objects if obj["shape"] == "sphere"
        ↪   and obj["material"] == "metal" and obj["size"] == "small" and obj["x"] >
        ↪   closest_big_ball["x"]), None)
36
37      return tiny_shiny_object["color"]
```

Figure B.2: Final symbols and generated program from PIPS for the instance shown in Figure 4b. The code returns "brown" which is the correct answer.

Table C.2: Accuracy and non-trivial-code percentage for o4-mini. Best accuracy per row is bolded.

| Dataset | CoT | PoT | PoT Non-Trivial (%) | PIPS | PIPS Non-Trivial (%) |
|---|---|---|---|---|---|
| Buggy tables | 0.275 | 0.512 | 85.6% | **0.594** | 96.9% |
| Temporal sequence | 0.325 | 0.306 | 92.5% | **0.519** | 96.2% |
| Dyck languages | **0.650** | 0.637 | 8.8% | 0.606 | 88.8% |
| Multistep arithmetic | 0.281 | 0.600 | 72.5% | **0.631** | 87.5% |
| Time arithmetic | 0.875 | **0.900** | 76.2% | 0.894 | 87.5% |
| Shuffled objects | 0.081 | 0.150 | 43.1% | **0.344** | 71.2% |
| Web of lies | 0.388 | 0.344 | 68.1% | **0.525** | 70.6% |
| Object counting | 0.850 | 0.812 | 93.1% | **0.900** | 70.0% |
| Zebra puzzles | 0.150 | 0.100 | 62.5% | **0.231** | 68.1% |
| Object properties | 0.144 | 0.219 | 38.8% | **0.344** | 65.6% |
| Boolean expressions | **0.550** | 0.469 | 24.4% | 0.419 | 63.7% |
| Spatial reasoning | 0.463 | 0.506 | 60.6% | **0.550** | 55.0% |
| Word sorting | **0.806** | 0.775 | 46.2% | **0.806** | 53.1% |
| Omnimath-2 | **0.812** | 0.706 | 56.9% | 0.787 | 29.4% |
| Movie recommendation | **0.819** | 0.744 | 10.6% | 0.731 | 18.8% |
| Omnimath-1 | 0.925 | 0.925 | 78.1% | **0.944** | 15.6% |
| Boardgame qa | **0.688** | 0.637 | 10.6% | 0.675 | 13.8% |
| Hyperbaton | **0.206** | 0.194 | 34.4% | 0.188 | 13.8% |
| Omnimath-3 | **0.556** | 0.356 | 36.9% | 0.544 | 11.9% |
| Omnimath-4 | **0.662** | 0.419 | 31.2% | 0.644 | 6.9% |
| Clutrr | 0.762 | **0.800** | 1.2% | 0.762 | 2.5% |
| Sportqa | **0.287** | 0.256 | 0.0% | **0.287** | 1.9% |
| Linguini | 0.138 | **0.175** | 6.2% | 0.138 | 1.2% |
| CLEVR | **0.769** | 0.750 | 6.2% | **0.769** | 0.6% |
| Causal understanding | **0.581** | 0.550 | 0.6% | **0.581** | 0.6% |
| Leaf | 0.364 | **0.455** | 0.0% | 0.364 | 0.0% |
| Geometric shapes | 0.056 | **0.119** | 0.0% | 0.087 | 0.0% |
| Disambiguation qa | 0.562 | **0.573** | 0.0% | 0.562 | 0.0% |
| Sarc triples | **0.338** | 0.300 | 0.0% | **0.338** | 0.0% |
| Nycc | **0.231** | 0.150 | 0.0% | **0.231** | 0.0% |
| Harmonic Mean | 0.304 | 0.340 | 0.0% | 0.397 | 0.1% |

compared to PoT or CoT, but PIPS is overall only 3-4X more expensive than CoT or PoT while achieving much greater accuracy.

## F Non-Trivial Program Synthesis on Non-Algorithmic Problems

We find that encouraging non-trivial programs for non-algorithmic problems leads to reduced performance. For instance, CoT with Gemini-2.0-Flash results in a harmonic mean accuracy over all non-algorithmic datasets (as listed in Table A.1) of 0.199 while PoT results in a lower value of 0.166 and our method without switching results in 0.151. Producing non-trivial programs for non-algorithmic problems which shouldn't be solved via code in the first place, harms performance. Therefore, a high performing general reasoning system needs to avoid program synthesis in such cases.

## G Program Evaluation Criteria

The eight criteria we use to evaluate code within PIPS are included below. The input dependence criteria is meant to catch when the program is trivial, the proper output criteria catches cases where the program does not output the answer in the correct format, and we also include the symbol extraction issues criteria to find issues during the first step of symbol extraction.

- Input dependence: Does the code use the input symbols to compute the answer?
- Valid return: Does the code avoid returning None unless it is the correct answer?

Table C.3: Accuracy and non-trivial-code percentage for Gemini-2.0-Flash. Best accuracy per row is bolded.

| Dataset | CoT | PoT | PoT Non-Trivial (%) | CI | PIPS | PIPS Non-Trivial (%) |
|---|---|---|---|---|---|---|
| Shuffled objects | 0.094 | 0.025 | 35.6% | **0.537** | 0.188 | 98.1% |
| Buggy tables | 0.019 | 0.100 | 99.4% | 0.031 | **0.188** | 97.5% |
| Time arithmetic | 0.438 | 0.331 | 82.5% | 0.312 | **0.475** | 97.5% |
| Temporal sequence | 0.006 | 0.006 | 42.5% | 0.006 | **0.094** | 96.2% |
| Multistep arithmetic | **0.144** | 0.037 | 87.5% | 0.087 | 0.119 | 90.6% |
| Dyck languages | **0.119** | 0.081 | 1.9% | 0.106 | 0.050 | 90.0% |
| Boolean expressions | 0.294 | 0.219 | 65.6% | 0.325 | **0.456** | 86.2% |
| Object counting | 0.144 | 0.119 | 98.1% | 0.181 | **0.281** | 85.0% |
| Omnimath-1 | 0.850 | 0.838 | 57.5% | 0.844 | **0.869** | 84.4% |
| Word sorting | 0.287 | 0.525 | 48.8% | 0.338 | **0.556** | 73.1% |
| Object properties | 0.006 | 0.062 | 26.9% | 0.125 | **0.163** | 71.9% |
| Spatial reasoning | 0.231 | **0.237** | 12.5% | 0.219 | 0.231 | 70.6% |
| CLEVR | 0.637 | 0.619 | 26.2% | 0.669 | **0.688** | 46.2% |
| Causal understanding | 0.537 | 0.438 | 1.2% | **0.544** | 0.537 | 15.6% |
| Clutrr | 0.556 | 0.588 | 0.0% | **0.662** | 0.506 | 13.8% |
| Linguini | **0.144** | 0.113 | 1.9% | 0.119 | 0.125 | 13.8% |
| Boardgame qa | **0.463** | 0.419 | 5.0% | 0.394 | **0.463** | 11.9% |
| Zebra puzzles | **0.300** | 0.256 | 73.8% | 0.131 | 0.275 | 10.0% |
| Geometric shapes | 0.312 | 0.269 | 0.6% | **0.388** | 0.300 | 9.4% |
| Sportqa | 0.200 | 0.244 | 1.9% | **0.269** | 0.194 | 6.9% |
| Hyperbaton | **0.031** | 0.019 | 46.2% | **0.031** | 0.025 | 5.6% |
| Web of lies | **0.219** | 0.206 | 3.1% | 0.188 | **0.219** | 2.5% |
| Movie recommendation | **0.581** | 0.562 | 0.0% | 0.556 | 0.569 | 2.5% |
| Disambiguation qa | 0.448 | 0.417 | 0.0% | **0.479** | 0.448 | 1.0% |
| Leaf | 0.602 | **0.636** | 0.0% | 0.102 | 0.602 | 0.0% |
| Sarc triples | **0.375** | 0.369 | 0.0% | 0.344 | **0.375** | 0.0% |
| Nycc | 0.106 | **0.131** | 0.0% | 0.113 | 0.106 | 0.0% |
| Omnimath-2 | **0.544** | 0.463 | 46.9% | **0.544** | **0.544** | 0.0% |
| Omnimath-3 | 0.269 | 0.194 | 15.6% | **0.275** | 0.269 | 0.0% |
| Omnimath-4 | 0.312 | 0.244 | 16.2% | **0.319** | 0.312 | 0.0% |
| Harmonic Mean | 0.107 | 0.115 | 0.0% | 0.134 | 0.201 | 0.0% |

- Proper output: Does the code return (not print) the correct answer in the expected format?
- No example usage: Does the code omit example calls or usage?
- Simplifiability: Could the solution be implemented in a simpler way?
- Correctness bugs: Are there any bugs affecting correctness?
- Symbol extraction issues: Are there any problems with the extracted input symbols?
- Sanity check: Does the output pass a basic sanity check?

# H  Switch Analysis

In this section, we go in-depth on the CoT vs. program synthesis switch design. First we discuss some additional evaluations involving evaluating the generalizability of the trained switch as well as evaluating a zero-shot switch.

## H.1  Switch Ablations

We perform ablations for the switch in PIPS and show the results in Table H.11. First, we show that if there is no calibration data available to train the switch, it can be used in a zero-shot manner and still be highly performant. The zero-shot switch uses only the last of the ten questions as the final classifier value so that no training is required.

To validate that training a classifier on any data (even from a different dataset) can still be useful, we perform a leave-one-dataset-out evaluation. This involves training the switch on all but one dataset

Table C.4: Accuracy and non-trivial-code percentage for gpt-4.1-mini-2025-04-14. Best accuracy per row is bolded.

| Dataset | CoT | PoT | PoT Non-Trivial (%) | PIPS | PIPS Non-Trivial (%) |
|---|---|---|---|---|---|
| Time arithmetic | 0.588 | **0.706** | 93.1% | 0.463 | 98.8% |
| Buggy tables | 0.075 | **0.481** | 100.0% | 0.406 | 98.1% |
| Multistep arithmetic | 0.275 | 0.394 | 86.2% | **0.494** | 96.2% |
| Dyck languages | 0.150 | 0.175 | 6.9% | **0.506** | 95.0% |
| Shuffled objects | 0.119 | 0.256 | 68.8% | **0.294** | 95.0% |
| Omnimath-1 | **0.894** | 0.875 | 76.2% | 0.875 | 89.4% |
| Word sorting | **0.681** | 0.600 | 73.1% | 0.656 | 85.6% |
| Object counting | 0.263 | **0.331** | 66.2% | 0.287 | 83.1% |
| Temporal sequence | 0.250 | **0.356** | 96.2% | 0.275 | 82.5% |
| Boolean expressions | 0.294 | 0.356 | 11.2% | **0.469** | 79.4% |
| Spatial reasoning | 0.181 | **0.394** | 48.1% | 0.362 | 68.1% |
| Object properties | 0.025 | **0.237** | 67.5% | 0.175 | 60.0% |
| Omnimath-2 | **0.569** | **0.569** | 63.1% | 0.556 | 59.4% |
| Web of lies | **0.362** | 0.300 | 36.2% | 0.219 | 58.8% |
| CLEVR | **0.719** | 0.669 | 8.1% | 0.700 | 53.8% |
| Zebra puzzles | **0.194** | 0.150 | 36.2% | 0.188 | 49.4% |
| Clutrr | **0.662** | 0.562 | 0.6% | 0.613 | 21.9% |
| Omnimath-3 | **0.338** | 0.244 | 35.6% | 0.325 | 16.2% |
| Geometric shapes | **0.344** | 0.294 | 16.9% | 0.319 | 15.0% |
| Boardgame qa | 0.512 | 0.500 | 81.2% | **0.519** | 11.2% |
| Omnimath-4 | **0.463** | 0.312 | 28.7% | 0.431 | 10.0% |
| Sportqa | 0.169 | **0.244** | 2.5% | 0.200 | 9.4% |
| Hyperbaton | **0.087** | 0.062 | 4.4% | 0.075 | 7.5% |
| Causal understanding | **0.562** | **0.562** | 1.2% | 0.550 | 5.6% |
| Linguini | 0.094 | **0.144** | 1.2% | 0.094 | 3.1% |
| Movie recommendation | **0.606** | 0.475 | 8.8% | 0.594 | 1.9% |
| Leaf | 0.341 | **0.409** | 0.0% | 0.341 | 0.0% |
| Disambiguation qa | **0.552** | 0.500 | 1.0% | **0.552** | 0.0% |
| Sarc triples | 0.287 | **0.331** | 10.0% | 0.287 | 0.0% |
| Nycc | **0.188** | 0.150 | 15.0% | **0.188** | 0.0% |
| Harmonic Mean | 0.211 | 0.297 | 0.3% | 0.305 | 0.1% |

and then evaluating PIPS over only the left out dataset and averaging performance over all datasets as the left out one. As shown in Table H.11, this also performs nearly as well as PIPS where the switch is trained over calibration set of data sampled from all datasets.

## H.2   Switch Criteria

The full prompt for the switch is provided in Appendix I, but we provide the 10 criteria we use within the prompt below. The criteria for determining if an instance should be solved directly via CoT or by program synthesis are the following:

1. Simple formalizability: Likelihood that the solution can be easily expressed as simple, deterministic code.

2. Straightforward executability: Likelihood that a first code attempt runs correctly without debugging.

3. Robust systematic search: Likelihood that systematic code (e.g., brute-force, recursion) reliably solves the problem.

4. Manageable state representation: Likelihood that all necessary variables and concepts can be cleanly represented in code.

5. Structured knowledge encoding: Likelihood that required background knowledge can be encoded as rules or data.

Table C.5: Accuracy for Qwen3-235B-A22B over all BBEH tasks. Best accuracy per row is bolded.

| Dataset | CoT | PoT | PIPS |
|---|---|---|---|
| Word sorting | **0.738** | 0.688 | 0.706 |
| Dyck languages | 0.438 | 0.200 | **0.456** |
| Object counting | **0.644** | 0.519 | 0.613 |
| Object properties | 0.319 | **0.475** | 0.250 |
| Boardgame qa | 0.744 | **0.838** | 0.744 |
| Boolean expressions | 0.600 | 0.362 | **0.631** |
| Buggy tables | 0.181 | **0.362** | 0.312 |
| Spatial reasoning | 0.512 | 0.506 | **0.519** |
| Multistep arithmetic | **0.550** | 0.544 | 0.275 |
| Geometric shapes | **0.237** | 0.200 | **0.237** |
| Temporal sequence | 0.294 | 0.331 | **0.350** |
| Disambiguation qa | **0.625** | 0.573 | **0.625** |
| Causal understanding | **0.569** | 0.456 | **0.569** |
| Time arithmetic | 0.613 | **0.750** | 0.700 |
| Web of lies | **0.812** | 0.631 | **0.812** |
| Sarc triples | **0.281** | 0.194 | **0.281** |
| Hyperbaton | **0.287** | **0.287** | **0.287** |
| Nycc | 0.156 | **0.163** | 0.156 |
| Sportqa | **0.256** | 0.150 | **0.256** |
| Linguini | **0.175** | 0.150 | **0.175** |
| Movie recommendation | **0.738** | 0.719 | **0.738** |
| Shuffled objects | 0.144 | 0.056 | **0.569** |
| Zebra puzzles | **0.700** | 0.569 | **0.700** |
| Harmonic Mean | 0.355 | 0.289 | **0.386** |

Table C.6: Harmonic Mean Accuracy (All Datasets)

| Method | gpt-4.1-mini | Gemini-2.0-Flash | o4-mini |
|---|---|---|---|
| CoT | 0.211 | 0.107 | 0.304 |
| PoT | 0.297 | 0.115 | 0.340 |
| PIPS | 0.305 | 0.201 | 0.397 |
| CI | 0.000 | 0.134 | 0.000 |

6. Hallucination risk reduction: Likelihood that code avoids fabricated steps better than chain-of-thought reasoning.

7. Arithmetic and data processing advantage: Likelihood that code handles arithmetic or data processing more reliably.

8. Branching and case handling advantage: Likelihood that code handles special cases or branching logic more systematically.

9. Algorithmic reliability over heuristics: Likelihood that a deterministic algorithm outperforms intuitive reasoning.

10. Overall comparative success: Likelihood that code yields a more reliable solution than chain-of-thought reasoning.

Our full prompt asks the LLM itself to quantify each of these criteria and then we build a simple logistic classifier based on the LLM's own judgements to determine when to use program synthesis.

## H.3    Examples of Switch Decisions

We show two questions from BBEH Word Sorting in Figure H.3 and the corresponding switch decisions for the different models. We also show a question from Omnimath-2 in Figure H.4 where the different CoT/Synthesis switch decisions are made for different models.

Table C.7: Harmonic Mean Accuracy (Algorithmic Datasets)

| Method | gpt-4.1-mini | Gemini-2.0-Flash | o4-mini |
|--------|--------------|------------------|---------|
| CoT    | 0.187        | 0.079            | 0.339   |
| PoT    | 0.357        | 0.094            | 0.389   |
| PIPS   | 0.369        | 0.217            | 0.548   |
| CI     | -            | 0.112            | -       |

Table C.8: Harmonic Mean Accuracy (Non-Algorithmic Datasets)

| Method | gpt-4.1-mini | Gemini-2.0-Flash | o4-mini |
|--------|--------------|------------------|---------|
| CoT    | 0.254        | 0.199            | 0.267   |
| PoT    | 0.244        | 0.166            | 0.291   |
| PIPS   | 0.248        | 0.184            | 0.292   |
| CI     | -            | 0.181            | -       |

## H.4 Question Analysis

We include the logistic regression weights for each of the ten questions in Appendix H.2 in Table H.12. The most important questions for the switch actually vary significantly between models. For Gemini-2.0-Flash, question 5 and 6 are the most important while for GPT-4.1-mini it is question 8 and 5. Finally, for o4-mini, questions 2 and 1 are the most important. Interestingly, we see that questions 5, 6, and 8 which are important for the non-reasoning models are related to the ability of the model to produce error-free code and avoid tedious steps using code while the reasoning model relies most on the criteria which concerns traditional problem algorithmicity rather than model capability.

## I Prompts

All prompts used in our evaluation and method are included below.

The prompt used to create the LLM-based classifier for question algorithmicity is the following.

---

**Algorithmic Question Evaluation**

```
You will determine whether a given target question can be definitively solved by↩
    writing a Python program (algorithmic) or if it necessitates another form ↩
    of reasoning (non-algorithmic). A Python solution may import standard ↩
    libraries, but cannot simply invoke external services, APIs, or LLMs.
If the input contains images, an algorithmic solution may use information ↩
    manually extracted without needing to interpret the image itself.

Evaluate the target question carefully against the following criteria. Answer ↩
    each sub-question rigorously with a binary response (1 for yes, 0 for no), ↩
    ensuring a high threshold for certainty:

1. Does the problem have explicitly defined inputs and outputs, such that ↩
    identical inputs always yield identical outputs?
2. Are there explicit, clearly stated rules, formulas, algorithms, or known ↩
    deterministic procedures available for solving this problem?
3. Does solving this problem strictly require exact computation (no ↩
    approximations, intuition, or interpretation)?
4. Can this problem be fully formalized in clear mathematical, logical, or ↩
    structured computational terms without ambiguity?
5. Can the solution method be decomposed into a finite, clear, and unambiguous ↩
    sequence of computational steps?
6. Is there a universally recognized and objective standard for verifying the ↩
    correctness of the solution?
7. Does solving the problem inherently involve repetitive or iterative ↩
    computations clearly suitable for automation?
8. Are the inputs structured, quantifiable, and inherently suited to algorithmic↩
    manipulation?
9. Does this problem clearly match or closely resemble a known, standardized ↩
    computational task or problem type?
```

---

Table D.9: Harmonic mean accuracy computed over BBEH tasks for additional agentic baselines compared to PIPS on Gemini-2.0-Flash. The highest accuracy is bolded.

| Method | HMean Accuracy |
|---|---|
| PoT | 0.095 |
| PoT-retries | 0.098 |
| CodeAct | 0.040 |
| BoT | 0.027 |
| PIPS | **0.171** |

Table E.10: Comparison of average token usage and cost across methods.

| Method | Avg. Input Tokens | Avg. Output Tokens | Cost (USD) |
|---|---|---|---|
| PoT | 1,115.96 | 1,333.98 | $0.0006 |
| CoT | 1,099.77 | 1,475.87 | $0.0007 |
| CodeAct | 80,137.92 | 4,023.36 | $0.0096 |
| Buffer of Thoughts | 340,927.19 | 123,655.35 | $0.0835 |
| PIPS | 11,839.09 | 2,805.78 | $0.0023 |

```
10. Is absolute correctness required (i.e., no margin for error or subjective ←
    interpretation)?

After thoroughly reasoning through these sub-questions, append a final ←
    determination as the 11th element:
- Output 1 if and only if all or nearly all (at least 8 out of 10) answers are ←
    clearly 1, indicating the problem is definitively algorithmic.
- Otherwise, output 0, indicating the problem requires non-algorithmic reasoning←
    .

IMPORTANT:
- Before providing the binary list, explicitly reason through each criterion ←
    carefully and thoroughly, clearly justifying your decisions. If uncertain ←
    or ambiguous about any criterion, default to 0.
- Provide your final answer explicitly as an 11-element binary list (ten answers←
     plus the final determination).
- Under no circumstances should you attempt to answer the actual target question←
     itself.

TARGET QUESTION:
```

The prompt used to extract the ten criteria for building our instance-level CoT or program synthesis switch is provided next.

**CoT or Synthesis Switch Criteria**

```
You will self-reflect to estimate whether you are more likely to correctly solve←
     a given target question by writing executable Python code or by using ←
    chain-of-thought (natural-language) reasoning.

**IMPORTANT:**
- This is a hypothetical evaluation.
- **You must NOT attempt to answer, solve, write code, or reason through the ←
    target question yet.**
- Instead, you must reflect carefully and conservatively on your expected ←
    ability if you were to attempt solving the question through either method.

Solution Expectations:
- You may assume standard library modules are allowed for code.
- You may NOT call external services, APIs, databases, or other LLMs.
- The code must be self-contained and executable without internet access.
- Chain-of-thought reasoning must be clear, logically sound, and internally ←
    verifiable without external tools.

**CRITICAL GUIDANCE:**
```

Table H.11: Ablations for the switch in PIPS compared using harmonic mean accuracy over all 30 datasets. The ZS-Switch uses only the model output from the final of the ten criteria as the final switch decision the leave-one-out (LOO) setting trains the switch over all but one dataset and then evaluates on the left out dataset and averages over all datasets being the left out one.

| Models | PIPS w. ZS-Switch | PIPS w. LOO | PIPS |
|---|---|---|---|
| o4-mini | 0.389 | 0.393 | 0.397 |
| Gemini-2.0-Flash | 0.199 | 0.180 | 0.208 |
| GPT-4.1-mini | 0.259 | 0.302 | 0.305 |

### Q1

You are an expert in word sorting. You will be provided with a list of words and the steps taken, in the form of thoughts, to arrange them in alphabetical order. Your job is to identify the first step that was a mistake in reasoning about the order of the words. This can be misunderstanding the order of the alphabet or getting the word or letter wrong or missing a word entirely. Q: Sort the following words alphabetically: List: electron amputate morphism tire scald lawmen putdown fishpond expositor yosemite gourmet transplant noon structure protagonist arab agree daisy delta eskimo Thought 1: I should start by looking at the first letter of the words in the list. The first letter: "electron": "e" (5). "amputate": "a" (1). "morphism": "m" (13). "tire": "t" (20). "scald": "s" (19). "lawmen": "l" (12). "putdown": "p" (16). "fishpond": "f" (6). "expositor": "e" (5). "yosemite": "y" (25). "gourmet": "g" (7). "transplant": "t" (20). "noon": "n" (14). "structure": "s" (19). "protagonist": "p" (16). "arab": "a" (1). "agree": "a" (1). "daisy": "d" (4). "delta": "d" (4). "eskimo": "e" (5).
...
Thought 15: I have now sorted all the words. The answer is agree amputate arab delta daisy electron expositor eskimo fishpond gourmet lawmen morphism noon protagonist putdown scald structure tire transplant yosemite Q: Is there a mistake in this sequence? Write "No" if there are no mistakes, or the number N if there is a mistake in Thought N.

### Q2

Consider a new alphabet whose letters have the same order as the English alphabet, except that c and m are the last two letters. Sort the following words with the new alphabet and separate them with comma: medea, oversimplifications, clonic, chaplin, kennan, postpone, squabble, ipsilateral, misunderstandings, ussr, canal, modifications, referring, counterrevolutionaries, pyridine, cameroon, avalanche, rationalizations, fortran, cram, coachman

Figure H.3: Two questions from the BBEH Word Sorting task where the first question asks for determining which thought in a model's CoT is incorrect and the second question asks for sorting a list of words. The switch for these problems chooses to answer Q1 with CoT for all models while answer Q2 with program synthesis for all models.

```
- **Be cautious, not optimistic.**
  Overestimating your capabilities will lead to choosing a method you cannot ←
      successfully complete.
- **If you feel any uncertainty, complexity, or ambiguity, lower your ←
    probability accordingly.**
- **Assume that even small mistakes can cause failure** when writing code or ←
    reasoning through complex tasks.
- **Use conservative estimates.**
- If unsure between two options, **prefer lower probabilities rather than ←
    guessing high**.

Here are the self-reflection sub-questions you must answer hypothetically:

1. **Simple Formalizability** - *What is the probability that the full solution ←
    can be easily and directly expressed as simple, deterministic code, without←
     needing complex transformations or deep insight?*

2. **Straightforward Executability** - *What is the probability that a first ←
    attempt at writing code would execute correctly without needing debugging, ←
    even if the problem has subtle or complex aspects?*

3. **Robust Systematic Search** - *What is the probability that coding a ←
    systematic method (like brute-force search or recursion) would reliably ←
```

**Q1**

A collection $\mathcal{S}$ of 10000 points is formed by picking each point uniformly at random inside a circle of radius 1. Let $N$ be the expected number of points of $\mathcal{S}$ which are vertices of the convex hull of the $\mathcal{S}$. (The convex hull is the smallest convex polygon containing every point of $\mathcal{S}$.) Estimate $N$.

Figure H.4: A question from Omnimath-2 where both Gemini-2.0-Flash and GPT-4.1-mini switch to CoT to answer while o4-mini chooses program synthesis.

Table H.12: Logistic regression coefficients for the switch for each of the three models.

| Model | Q1 | Q2 | Q3 | Q4 | Q5 | Q6 | Q7 | Q8 | Q9 | Q10 |
|---|---|---|---|---|---|---|---|---|---|---|
| Gemini-2.0-Flash | 0.14 | 0.03 | 0.12 | 0.15 | 0.21 | -0.21 | 0.18 | -0.09 | 0.03 | 0.10 |
| gpt-4.1-mini | 0.40 | 0.42 | 0.20 | 0.28 | 0.22 | 0.15 | -0.02 | 0.01 | 0.21 | 0.21 |
| o4-mini | 0.22 | 0.04 | 0.16 | 0.24 | 0.27 | -0.05 | 0.12 | 0.35 | 0.18 | 0.24 |

```
       find the correct answer, without missing hidden constraints or introducing ↵
          edge-case errors?*

4. **Manageable State Representation** - *What is the probability that all ↵
      intermediate concepts, variables, and conditions can be simply and ↵
      explicitly represented in code, without requiring difficult or error-prone ↵
      state tracking?*

5. **Structured Knowledge Encoding** - *What is the probability that all ↵
      required background knowledge can be neatly encoded in code (e.g., as rules↵
      , formulas, or data), rather than needing flexible, intuitive understanding↵
       better suited to reasoning?*

6. **Hallucination Risk Reduction** - *What is the probability that code ↵
      execution would more reliably avoid fabricated steps or unwarranted ↵
      assumptions compared to chain-of-thought reasoning?*

7. **Arithmetic and Data Processing Advantage** - *What is the probability that ↵
      the problem requires extensive or error-prone arithmetic/data handling that↵
       code could perform perfectly, but that chain-of-thought would likely ↵
      fumble?*

8. **Branching and Case Handling Advantage** - *What is the probability that the↵
       solution involves many branching conditions, special cases, or exceptions ↵
      that code can handle systematically but chain-of-thought might overlook?*

9. **Algorithmic Reliability Over Heuristics** - *What is the probability that ↵
      following a deterministic algorithm in code would reach the correct answer ↵
      more reliably than relying on intuitive or heuristic chain-of-thought ↵
      reasoning?*

10. **Overall Comparative Success** - *Considering all factors, what is the ↵
      probability that code will ultimately produce a correct solution more ↵
      reliably than chain-of-thought reasoning for this question?*

After thoroughly reasoning through each criterion:

- Output a single list of 10 probability scores (each between 0 and 1) as your ↵
      FINAL ANSWER, in order:
  - Scores 1-10 correspond to the ten sub-questions above.

**Additional Instructions:**
- Explicitly reason through each criterion carefully before giving a probability↵
      .
- If uncertain or if the problem seems complex, favor lower probabilities to ↵
      reflect the difficulty.
- Make sure to put only the list after FINAL ANSWER.
- **Under no circumstances should you write, sketch, pseudocode, or attempt any ↵
      part of the solution itself during this reflection phase.**

TARGET QUESTION:
```

The prompt for generating the first program with PIPS in iteration one is the following.

**PIPS Code Generator (Iteration 0)**

```
You will be given a question and you must answer it by extracting relevant ←
    symbols in JSON format and then writing a Python program to calculate the ←
    final answer.

You MUST always plan extensively before outputting any symbols or code.

You MUST iterate and keep going until the problem is solved.

# Workflow

## Problem Solving Steps
1. First extract relevant information from the input as JSON. Try to represent ←
    the relevant information in as much of a structured format as possible to ←
    help with further reasoning/processing.
2. Using the information extracted, determine a reasonable approach to solving ←
    the problem using code, such that executing the code will return the final ←
    answer.
3. Write a Python program to calculate and return the final answer. Use comments←
    to explain the structure of the code and do not use a main() function.
The JSON must be enclosed in a markdown code block and the Python function must ←
    be in a separate markdown code block and be called `solve` and accept a ←
    single input called `symbols` representing the JSON information extracted. ←
    Do not include any `if __name__ == "__main__"` statement and you can assume←
    the JSON will be loaded into the variable called `symbols` by the user.
The Python code should not just return the answer or perform all reasoning in ←
    comments and instead leverage the code itself to perform the reasoning.
Be careful that the code returns the answer as expected by the question, for ←
    instance, if the question is multiple choice, the code must return the ←
    choice as described in the question.
Be sure to always output a JSON code block and a Python code block.
```

The prompt used to generate subsequent code solutions with PIPS by leveraging the evaluator output is shown below.

**PIPS Code Generator (Iteration > 0)**

```
Please fix the issues with the code and symbols or output "FINISHED".
The following is the result of evaluating the above code with the extracted ←
    symbols.
```
Return value: {output}
Standard output: {stdout}
Exceptions: {err}
```

The following is the summary of issues found with the code or the extracted ←
    symbols by another model:
```
{checker_output}
```

If there are any issues which impact the correctness of the answer, please ←
    output code which does not have the issues. Before outputting any code, ←
    plan how the code will solve the problem and avoid the issues.
If stuck, try outputting different code to solve the problem in a different way.
You may also revise the extracted symbols. To do this, output the revised ←
    symbols in a JSON code block. Only include information in the JSON which is←
    present in the original input to keep the code grounded in the specific ←
    problem. Some examples of symbol revisions are changing the names of ←
    certain symbols, providing further granularity, and adding information ←
    which was originally missed.
If everything is correct, output the word "FINISHED" and nothing else.
```

Finally, the prompt for the code evaluator is shown below.

**PIPS Evaluator**

```
You will be given a question and a code solution and you must judge the quality ↩
    of the code for solving the problem.

Look for any of the following issues in the code:
- The code should be input dependent, meaning it should use the input symbols to↩
    compute the answer. It is OK for the code to be specialized to the input (↩
    i.e. the reasoning itself may be hardcoded, like a decision tree where the ↩
    branches are hardcoded).
- The code should not return None unless "None" is the correct answer.
- The code should return the answer, not just print it. If the question asks for↩
    a multiple choice answer, the code should return the choice as described ↩
    in the question.
- There should not be any example usage of the code.
- If there is a simpler way to solve the problem, please describe it.
- If there are any clear bugs in the code which impact the correctness of the ↩
    answer, please describe them.
- If there are any issues with the extracted symbols, please describe them as ↩
    well, but separate these issues from the issues with the code.
- If it is possible to sanity check the output of the code, please do so and ↩
    describe if there are any obvious issues with the output and how the code ↩
    could be fixed to avoid these issues.

After analyzing the code in depth, output a concrete and concise summary of the ↩
    issues that are present, do not include any code examples. Please order the↩
    issues by impact on answer correctness.
The following are extracted symbols from the question in JSON format followed by↩
    a Python program which takes the JSON as an argument called `symbols` and ↩
    computes the answer.
```json
{json_str}
```

```python
{code_str}
```

Code execution result:
```
Return value: {output}
Standard output: {stdout}
Exceptions: {err}
```

Output a concrete and concise summary of only the issues that are present, do ↩
    not include any code examples.
```

## J  Hyperparameters

For all models we used a temperature of 0.0. For PIPS, we used a maximum of 30 iterations for all models.

## K  Compute Resources

For most experiments we rely on API model access. All experiments cost $300 for Gemini-2.0-Flash, $70 GPT-4.1-mini, and $700 for o4-mini (medium). Other experiments were run on a server with 96 Intel(R) Xeon(R) Gold 5318Y CPUs @ 2.10GHz with 1TB of system RAM. The server also had 10x NVIDIA A100 80GB GPUs which were only used for local testing of open-weights models.

## L  Broader Impacts

PIPS offers significant potential benefits, including enhanced AI reliability, trust, transparency, and the democratization of advanced problem-solving. However, like most systems built from powerful foundation models, it also presents important considerations. These include the potential for bias propagation from underlying models, challenges in ensuring the complexity and robustness of dynamically generated programs, and concerns regarding misuse. Continued research and development

must prioritize robust safeguards, fairness, transparency, and responsible deployment practices to harness the benefits while mitigating these potential negative impacts.

## M    Connection to Transductive Learning

Instance-wise Program Synthesis can be connected to transductive learning [46] where a function is learned to map from specific function inputs to specific outputs. The general philosophy is that one should not try to solve the general problem when the specific case is all one needs.

We are given a labeled training set $D_L = \{(x_i, y_i)\}_{i=1}^m \in \mathcal{R} \times \mathcal{Y}$ where $\mathcal{R}$ is the space of symbolic input and $\mathcal{Y}$ is the output space, and an unlabeled test sample $x \in \mathcal{R}$. Our goal is to find a program $p : \mathcal{R} \to \mathcal{Y}$ which has a low error on the given sample, so $p(x) = y$. In contrast, inductive program synthesis tries to find a program $p$ which *generalizes*, so it should achieve a low error on samples from the same distribution as $D_L$. To perform transductive program synthesis, we rely on minimizing an auxiliary "regularization" term, $\Omega$, over the test sample, representing a form of test-time computation:

$$\hat{p} = \arg \min_p \left\{ \frac{1}{m} \sum_{i=1}^m \ell(p(x_i), y_i) + \Omega(p; x_1, \ldots, x_m, x) \right\}.$$

The regularization term can be implemented using probabilistic priors (e.g. the likelihood under some language model) or more complex heuristics involving static/dynamic code analysis. With powerful foundation models that can perform zero-shot inference, we can even perform this type of learning without any training set.

**Neuro-symbolic Instance-level Synthesis**    The above problem definition is specifically for problems formulated for program synthesis, meaning the inputs are expressed in symbolic form. Can we apply such a method for problems expressed in natural language or even using images? To convert general problems into a form which is conducive to program execution, we use a Neuro-Symbolic framework, specifically a Prompt-Symbolic approach.

We now assume our samples come from an arbitrary raw input space, such as natural language, images, and other modalities. Using the traditional neuro-symbolic framework, we first extract symbols from the raw input and then pass these symbols to a program. Let $C : \mathcal{X} \to \mathcal{R}$ be a mapping from raw data to symbols. Now we adapt the above formalism as the following:

$$\hat{p}, \hat{C} = \arg \min_{p, C} \left\{ \frac{1}{m} \sum_{i=1}^m \ell(p(C(x_i)), y_i) + \Omega(C, p; x_1, \ldots, x_m, x) \right\}.$$

Intuitively, we want to find a raw data to symbols mapping $\hat{C}$ and program $\hat{p}$ which perform well on a training set (if one is available) and minimize the loss $\Omega$ over the given test samples. Notably, the program and raw data to symbol mapping do not need to be general, they should be *specialized* for the test samples. In practice, the raw data to symbol mapping is often a form of semantic parsing, which can be done through zero-shot prompting of foundation models, so the raw data to symbol mapping is not modified on an instance-level.

