# OpenReview forum: "Once Upon an Input: Reasoning via Per-Instance Program Synthesis"
_NeurIPS.cc/2025/Conference — NeurIPS 2025 poster_

### Official Review · Reviewer_7yLF · 2025-07-01

**Clarity:** 4
**Significance:** 2
**Originality:** 3
**Rating:** 4
**Confidence:** 3

**Summary:**

This work introduces Per-Instance Program Synthesis ("PIPS") which aims to increase accuracy on a variety of reasoning tasks.  Key elements include the detection of whether plain CoT is sufficient, or whether code should be written, with additional quality checks being applied to the code to ensure that it doesn't fall into a number of reasoning traps that the authors identified from other approaches.

**Questions:**

You state "our confidence metric allows us to correctly switch between CoT and program synthesis for 65% of cases" : As noted above, this seems as if it is a fairly weak classifier, somewhat undermining its significance.  Can I confirm that the results for the methods (other than PIPS) were also conducted on the 80% of the benchmark sets that were not held out as calibration data for the logistic classifier (L139) ?

Am I correct in suggesting that the PoT implementation did not allow for code retries/rewrites when there were errors, whereas PIPS was given the opportunity for multiple self-correction loops (as illustrated in Figure 6)?

**Ethical Concerns:**

["NO or VERY MINOR ethics concerns only"]

**Final Justification:**

I'm raising, but fairly reluctantly : The method clearly works, and the experimental work is fine (and improved through the review process).  OTOH, this is still fundamentally a tweak to chose between two styles of model question answering - finessing on an analysis of model errors, rather than adding a new capability into the mix

**Limitations:**

yes

**Quality:**

3

**Strengths And Weaknesses:**

One key element of PIPS seems to be the learned logistic classifier over features output by asking the LLM itself about its confidence in its own capabilities at answering the question with/without PoT.  While practical to implement, it doesn't seem as if this classifier achieves good accuracy in its determinations - which begs the question of whether there could be a simple probe (for a model with open weights, of course) that would perform better.

The investigation of failure modes for the code creation is of practical relevance - with the identification of 'failed structural properties' being a novel twist.

However, it seems that (from a quick look at the provided code) that the PoT implementation does not allow for regeneration/fixing of code errors, whereas PIPS is allowed re-tries.  This means that the PIPS method isn't purely benefiting from the structural property guardrails, but also from the ability to fix simple typo-style errors.  Choosing weak versions of the competitor baselines (as appears to be the case here, please correct me if I didn't find the correct code) isn't a good look.

---

> ### Author Rebuttal · Authors · 2025-07-31
>
> We thank the reviewer for the thoughtful comments. We address the reviewer’s specific concerns in detail below.
>
> ## Confidence Metric
>
> For our confidence-based switch between CoT and code synthesis, the 65% reported accuracy is only over the problems where exactly one of synthesis or CoT gets the correct answer, so this excludes cases where both are correct or both incorrect. Out of these problems where the outcome of CoT and synthesis differ, the best performing method out of the two gets 51.2% accuracy, so 65% accuracy is a significant improvement over both CoT and synthesis. There is still room for improvement in this switch (possibly through probe-based approaches as suggested by the reviewer), but we wanted an approach that would be easy to apply to new models, which to-date often do not provide hidden state access.
>
> ## Evaluation of Baselines on 80% Non-Heldout Data
>
> The reviewer is correct that we evaluated all methods on the 80% of the data that was not included in the heldout set used for calibration.
>
> ## Iterative Baselines
>
> The reviewer is correct that the PoT baseline did not allow for regeneration and fixing of code errors. The original version of PoT and other related methods such as FCoT [1] and PAL [2] also do not reprompt the model if there are execution errors. To address this concern, we re-run PoT with retries.   Addionally, in our experiments, we included the Gemini code interpreter as a baseline which can iteratively write and debug code, as mentioned on line 188. For another such baseline, we have added the smolagents library implementation of CodeAct [3], which we find performs poorly on BBEH with gemini-2.0-flash. We will include the CodeAct results for all models in the camera ready version as well as Buffer of Thoughts (a baseline suggested by another reviewer).
>
>
> | Dataset              | CodeAct (gemini-2.0-flash) |
> |----------------------|---------------------|
> | Word sorting         | 0.525               |
> | Dyck languages       | 0.044               |
> | Object counting      | 0.000               |
> | Object properties    | 0.006               |
> | Boardgame qa         | 0.325               |
> | Boolean expressions  | 0.256               |
> | Buggy tables         | 0.025               |
> | Spatial reasoning    | 0.287               |
> | Multistep arithmetic | 0.000               |
> | Geometric shapes     | 0.369               |
> | Temporal sequence    | 0.000               |
> | Disambiguation qa    | 0.469               |
> | Causal understanding | 0.512               |
> | Time arithmetic      | 0.256               |
> | Web of lies          | 0.206               |
> | Sarc triples         | 0.331               |
> | Hyperbaton           | 0.025               |
> | Nycc                 | 0.106               |
> | Sportqa              | 0.212               |
> | Linguini             | 0.125               |
> | Movie recommendation | 0.388               |
> | Shuffled objects     | 0.006               |
> | Zebra puzzles        | 0.031               |
> | Harmonic Mean        | 0.040               |
>
>
> | Dataset              | Buffer of Thoughts (gemini-2.0-flash) |
> |----------------------|--------------------|
> | Word sorting         | 0.463              |
> | Dyck languages       | 0.056              |
> | Object counting      | 0.006              |
> | Object properties    | 0.000              |
> | Boardgame qa         | 0.275              |
> | Boolean expressions  | 0.031              |
> | Buggy tables         | 0.000              |
> | Spatial reasoning    | 0.113              |
> | Multistep arithmetic | 0.000              |
> | Geometric shapes     | 0.006              |
> | Temporal sequence    | 0.000              |
> | Causal understanding | 0.362              |
> | Time arithmetic      | 0.100              |
> | Web of lies          | 0.087              |
> | Sarc triples         | 0.194              |
> | Hyperbaton           | 0.013              |
> | Nycc                 | 0.019              |
> | Sportqa              | 0.125              |
> | Linguini             | 0.006              |
> | Movie recommendation | 0.013              |
> | Zebra puzzles        | 0.069              |
> | Harmonic Mean        | 0.026              |
>
>
> Finally, we perform an additional experiment where we resample a potential solution using PoT whenever the generated code results in a code related error or a null result. The following shows the harmonic mean accuracy over all BBEH tasks using Gemini 2.0 Flash as the base model. We see that resampling is minimally beneficial by itself.
>
> | Methods            | Harmonic Mean Acc. |
> |--------------------|--------------------|
> | PoT                | 0.095              |
> | PoT (with retries) | 0.098              |
>
>
> [1] Lyu, Qing, et al. "Faithful chain-of-thought reasoning." IJCNLP-AACL 2023.
> [2] Gao, Luyu, et al. "Pal: Program-aided language models." ICML 2023.
> [3] Wang, Xingyao, et al. "Executable code actions elicit better llm agents." ICML 2024.

---

> > ### Comment · Reviewer_7yLF · 2025-08-04
> >
> > Thanks for running these additional experiments (even though they are reflective of what would have been a fair baseline comparison in the first place).
> >
> > Having read over all the reviews/rebuttals, though, I am inclined to leave my Rating unchanged.

---

> > > ### Author Response · Authors · 2025-08-07
> > >
> > > Thank you for the follow-up. We believe we addressed your concerns regarding fair baselines and the switch performance in the rebuttal. To summarize, the additions we will include in the revision:
> > > - **Iterative Baselines**: Our original submission already included the Gemini Code Interpreter (iterative). We will add PoT+retries (which yields only a modest gain over PoT; e.g., HM 0.095→0.098 on BBEH with gemini-2.0-flash) and report CodeAct and Buffer-of-Thoughts results for all models.
> > > - **Cost**: We will additionally include the cost results in the revision (copied below) which show that PIPS achieves the highest accuracy in three times less cost than CodeAct and 35 times less cost than Buffer of Thoughts. This shows that PIPS is more cost effective than other iterative baselines.
> > > | Method    | Avg. Input Tokens | Avg. Output Tokens | Cost (USD) |
> > > |-----------|-------------------|--------------------|------------|
> > > | PoT       | 1115.96           | 1333.98            | $0.0006    |
> > > | CoT       | 1099.77           | 1475.87            | $0.0007    |
> > > | CodeAct   | 80137.92          | 4023.36            | $0.0096    |
> > > | Buffer of Thoughts | 340927.19 | 123655.35 | $ 0.0835 |
> > > | PIPS      | 11839.09          | 2805.78            | $0.0023    |
> > > - **Switch Significance**: The reported 65% switch accuracy is computed only on instances where CoT and synthesis disagree. The best static choice on this subset is 51.2%, so our switch is a +13.8% absolute improvement.
> > >
> > > Thank you again for pointing out the important PoT+retries baseline which we believe has significantly strengthened our experiments. If there are any other baselines which the reviewer thinks should be added, we will be happy to include them as well.
> > >
> > > Regarding the other reviews/rebuttals, we believe we have addressed all the questions and concerns from other reviewers, but if there are any remaining concerns, we would appreciate the feedback and will try our best to respond.

---

### Official Review · Reviewer_JESL · 2025-07-02

**Clarity:** 2
**Significance:** 2
**Originality:** 2
**Rating:** 4
**Confidence:** 3

**Summary:**

The paper proposes Per-Instance Program Synthesis (PIPS), a test-time reasoning framework that decides, for each input, based on an instance-level confidence metric, whether to answer with plain chain-of-thought (CoT) reasoning or to synthesize and iteratively refine a program. Specifically, when code is chosen, the LLM/reasoner generates a symbolic abstraction of the raw input, writes a program that consumes that abstraction, runs it, and receives feedback on syntax, type, hard-coded answers and other failure patterns. Across three  LLMs (Gemini-2.0-Flash, GPT-4.1-mini, o4-mini) and 30 benchmarks (23 BBEH tasks plus CLEVR, Leaf, CLUTTR and four OmniMath tasks) PIPS lifts harmonic-mean accuracy by up to 8.6% over PoT and cuts undesirable programs by 65.1% .

**Questions:**

Please refer to the weaknesses listed in Strengths And Weaknesses and address all.

**Ethical Concerns:**

["NO or VERY MINOR ethics concerns only"]

**Final Justification:**

The author's additional experiments have addressed my concerns. I am therefore inclined to raise my rating to 4.

**Limitations:**

yes

**Quality:**

2

**Strengths And Weaknesses:**

## Strengths
1. The paper clearly articulates three core limitations of instance-level code generation: when to invoke code, absence of formal specifications, and unstructured natural language inputs. The proposed pipeline is well-aligned with these issues.
2. The approach is evaluated across diverse reasoning domains, such as language, vision-language, relational, and mathematical reasoning tasks, with comprehensive ablations on synthesis iterations, symbolic inputs, and code-switching heuristics.
3. The authors commit to releasing prompts and compute details. The main paper also includes a section for limitations and societal considerations, demonstrating transparency.
## Weaknesses
1. Both conceptually and empirically, authors mainly discuss the proposed approach in comparision with CoT and PoT, overlooking more recent and relevant works such as [1] (which enhances the effectiveness of solving multi-step reasoning problems by using code prompts as intermediate prompts) and [2] (which curates a library of high-level code templates (“thought-templates”) distilled from solved problems; the proposed approach retrieves & instantiates a template, then executes it).
2. All three LLMs used (o4-mini, Gemini-2.0-Flash, GPT-4.1-mini) are proprietary. The lack of open-source LLMs limits reproducibility and transparency.
3. Holding out 20% of every task for calibration may inflate reported accuracy and is unrealistic for truly zero-shot settings.
4. Efficiency aspect remains under-explored. Runtime overhead of iterative synthesis (plus interpreter calls) is not reported; Figure 6 shows accuracy vs iterations but not explicit numbers (e.g., token consumption and runtime).
5. While the paper acknowledges potential errors in converting inputs to structured symbolic form (e.g., JSON) in Section 3.3, it offers no quantitative assessment of semantic fidelity or its impact on downstream performance.
6. In Lines 195-196, the authors report that overall, "a 0.8% absolute improvement over PoT for GPT-4.1-mini". This improvement is trivial, and raises concerns about the approach’s diminishing returns as models improve. The paper should discuss this issue more explicitly.

[1] When Do Program-of-Thought Works for Reasoning? AAAI 2024.

[2] Buffer of Thoughts: Thought-Augmented Reasoning with Large Language Models. NeurIPS 2024 Spotlight.

---

> ### Author Rebuttal · Authors · 2025-07-31
>
> We thank the reviewer for the thoughtful comments. We address the reviewer’s specific concerns in detail below.
>
> ## Baselines
>
> While the paper [1] does not present a zero-shot method for LLM reasoning, we have added a comparison with Buffer of Thoughts [2] as suggested by the reviewer. This method uses other test problems from the same task to help with determining an effective method for solving other problems of the task. Overall, we find that this is highly ineffective for BBEH problems, performing even worse than just CoT or PoT alone.
>
> | Dataset              | Buffer of Thoughts (gemini-2.0-flash) |
> |----------------------|--------------------|
> | Word sorting         | 0.463              |
> | Dyck languages       | 0.056              |
> | Object counting      | 0.006              |
> | Object properties    | 0.000              |
> | Boardgame qa         | 0.275              |
> | Boolean expressions  | 0.031              |
> | Buggy tables         | 0.000              |
> | Spatial reasoning    | 0.113              |
> | Multistep arithmetic | 0.000              |
> | Geometric shapes     | 0.006              |
> | Temporal sequence    | 0.000              |
> | Causal understanding | 0.362              |
> | Time arithmetic      | 0.100              |
> | Web of lies          | 0.087              |
> | Sarc triples         | 0.194              |
> | Hyperbaton           | 0.013              |
> | Nycc                 | 0.019              |
> | Sportqa              | 0.125              |
> | Linguini             | 0.006              |
> | Movie recommendation | 0.013              |
> | Zebra puzzles        | 0.069              |
> | Harmonic Mean        | 0.026              |
>
>
> We note that we also included an agentic baseline in our original submission, the Gemini code interpreter, which we described on line 188. For another agentic baseline, we have included the smolagents implementation of a CodeAct agent, and we provide the results for all BBEH datasets using Gemini 2.0 Flash below. The complete results for all models will be included in the camera ready.
>
> | Dataset              | CodeAct (gemini-2.0-flash) |
> |----------------------|---------------------|
> | Word sorting         | 0.525               |
> | Dyck languages       | 0.044               |
> | Object counting      | 0.000               |
> | Object properties    | 0.006               |
> | Boardgame qa         | 0.325               |
> | Boolean expressions  | 0.256               |
> | Buggy tables         | 0.025               |
> | Spatial reasoning    | 0.287               |
> | Multistep arithmetic | 0.000               |
> | Geometric shapes     | 0.369               |
> | Temporal sequence    | 0.000               |
> | Disambiguation qa    | 0.469               |
> | Causal understanding | 0.512               |
> | Time arithmetic      | 0.256               |
> | Web of lies          | 0.206               |
> | Sarc triples         | 0.331               |
> | Hyperbaton           | 0.025               |
> | Nycc                 | 0.106               |
> | Sportqa              | 0.212               |
> | Linguini             | 0.125               |
> | Movie recommendation | 0.388               |
> | Shuffled objects     | 0.006               |
> | Zebra puzzles        | 0.031               |
> | Harmonic Mean        | 0.040               |
>
> [1] When Do Program-of-Thought Works for Reasoning? AAAI 2024.
> [2] Buffer of Thoughts: Thought-Augmented Reasoning with Large Language Models. NeurIPS 2024 Spotlight.
>
> ## LLMs Used
>
> Following the reviewer’s suggestion, we have evaluated the Qwen3-235B-A22B-Instruct-2507 model as an open-weights model. We report preliminary results over all of the BBEH tasks for this model and will include the complete results in the camera ready version. We observe a 1.3% absolute improvement in harmonic mean accuracy.
>
> | Dataset              | PoT   | CoT   | PIPS  |
> |----------------------|-------|-------|-------|
> | Word sorting         | 0.669 | 0.750 | 0.750 |
> | Dyck languages       | 0.169 | 0.419 | 0.456 |
> | Object counting      | 0.544 | 0.656 | 0.644 |
> | Object properties    | 0.475 | 0.338 | 0.331 |
> | Boardgame qa         | 0.844 | 0.738 | 0.738 |
> | Boolean expressions  | 0.325 | 0.600 | 0.594 |
> | Buggy tables         | 0.400 | 0.169 | 0.231 |
> | Spatial reasoning    | 0.550 | 0.525 | 0.525 |
> | Multistep arithmetic | 0.544 | 0.531 | 0.494 |
> | Geometric shapes     | 0.175 | 0.244 | 0.244 |
> | Temporal sequence    | 0.338 | 0.294 | 0.300 |
> | Causal understanding | 0.450 | 0.606 | 0.606 |
> | Time arithmetic      | 0.762 | 0.569 | 0.738 |
> | Web of lies          | 0.613 | 0.831 | 0.831 |
> | Sarc triples         | 0.181 | 0.287 | 0.287 |
> | Hyperbaton           | 0.256 | 0.287 | 0.287 |
> | Nycc                 | 0.169 | 0.131 | 0.131 |
> | Sportqa              | 0.163 | 0.281 | 0.281 |
> | Linguini             | 0.144 | 0.181 | 0.181 |
> | Movie recommendation | 0.750 | 0.725 | 0.725 |
> | Zebra puzzles        | 0.581 | 0.706 | 0.706 |
> | Harmonic Mean        | 0.326 | 0.366 | 0.379 |
>
>
> ## 20% Holdout Set for Calibration
>
> We thank the reviewer for bringing this up. We agree that for a strict zero-shot setting, the use of a validation set is not very standard. We only use the validation set for the switch and do not use any of the validation samples in any of the prompts or for improving the downstream CoT or synthesis methods. It is possible to implement the switch without any validation samples by just using the value of the tenth question rather than training a classifier to combine all ten questions. This zero-shot switch correctly chooses between CoT and synthesis 60.3% of the time for the problems where the outcome correctness of CoT and synthesis differs. Since choosing the method which performs the best between CoT and synthesis on these problems results in only 51.2% accuracy, this result is much better than both CoT or synthesis alone, but slightly worse than the trained switch which achieves 65% accuracy. We also evaluate a leave-one-out evaluation setting where the switch used on one dataset is only trained on the 20% validation samples from other datasets. This method, called “PIPS w. LOO” shows that the switch generalizes to held out datasets rather than just held out samples.
>
> We will include results for the setting with a switch implemented without any validation samples in our revised paper as well as the leave-one-out ablation. The PIPS w. LOO method still outperforms all baselines. For all models except GPT 4.1-mini (which already had the weakest performance), PIPS with a zero-shot switch still significantly outperforms the strongest baselines.
>
> | Models           | PIPS w. ZS Switch | PIPS w. LOO | PIPS  |
> |------------------|-------------------|-------------|-------|
> | o4-mini          | 0.389             | 0.393       | 0.397 |
> | Gemini 2.0 Flash | 0.199             | 0.180       | 0.208 |
> | GPT 4.1-mini     | 0.259             | 0.302       | 0.305 |
>
> ## Cost Overhead of PIPS
>
> We have calculated the average input and output tokens as well as average cost for each baseline method and PIPS for Gemini 2.0 Flash. We find that the iterative approaches (CodeAct and Buffer of Thoughts) increase cost over 10X compared to PoT or CoT, but PIPS is overall only 3-4X more expensive than CoT or PoT while achieving much greater accuracy.
>
> | Method    | Avg. Input Tokens | Avg. Output Tokens | Cost (USD) |
> |-----------|-------------------|--------------------|------------|
> | PoT       | 1115.96           | 1333.98            | $0.0006    |
> | CoT       | 1099.77           | 1475.87            | $0.0007    |
> | CodeAct   | 80137.92          | 4023.36            | $0.0096    |
> | Buffer of Thoughts | 340927.19 | 123655.35 | $ 0.0835 |
> | PIPS      | 11839.09          | 2805.78            | $0.0023    |
>
> ## Quantitative Assessment of Semantic Fidelity of Input to Symbolic Mapping
>
> Since our method first abstracts the raw input into a symbolic form, represented as JSON, there may be issues in this raw-to-symbolic mapping, as noted by the reviewer. Our preliminary investigation revealed that the vast majority of problems occur in the code itself (shown in Figure 3) and the model is often very good at the input to symbolic form translation. Now that we have addressed some of the most common problems, we think finer-grained issues should be studied in greater depth in future work, such as carrying out annotation efforts to determine the fidelity of the raw input to symbolic form mapping.
>
> ## Low Improvement for GPT 4.1-mini
>
> Our method results in only a 0.8% absolute increase in harmonic mean accuracy for GPT 4.1-mini, but this is our worst result out of many strong results across other models. In addition, we are adding an open-weights model, Qwen3-235B-A22B-Instruct-2507 (results are included above), where we see better results.
>
> We believe PIPS has a low effectiveness with GPT 4.1-mini because the model struggles at revising generated code based on the structural feedback we provide, due to its overall weak performance. For o4-mini, a model which performs better overall in terms of the baselines, we see that PIPS improves over PoT performance by 5.7% absolute harmonic mean accuracy. Therefore, we still see that stronger models significantly benefit from using PIPS.

---

> > ### Author Response · Authors · 2025-08-07
> >
> > Hi Reviewer JESL,
> >
> > Thanks again for your review. We believe we have addressed all your concerns and will include in our revision the two additional baselines (CodeAct and Buffer of Thoughts), an open-weights model, and the cost overhead information which we discussed in our rebuttal. If there are any remaining questions or concerns, we’d be grateful for your feedback.

---

> > ### Comment · Reviewer_JESL · 2025-08-07
> >
> > I appreciate the authors' efforts in conducting the additional experiments and will revise my score accordingly.
> >
> > I recommend that all additional results be incorporated into the paper.

---

### Official Review · Reviewer_ki2H · 2025-07-02

**Clarity:** 3
**Significance:** 2
**Originality:** 2
**Rating:** 4
**Confidence:** 3

**Summary:**

This paper introduces Per-Instance Program Synthesis (PIPS), a method designed to improve the multi-step reasoning abilities of large language models (LLMs). PIPS addresses the shortcomings of existing approaches like Program of Thought (PoT), which often produce undesirable solutions.The core of PIPS is to generate and refine programs at the individual instance level, guided by structural feedback rather than explicit test cases. The method includes two key innovations: A confidence metric that dynamically decides whether to solve a problem via program synthesis or direct inference on a per-instance basis;An iterative synthesis loop that uses an evaluator to check for structural flaws like hardcoded answers, type errors, or syntax issues, providing feedback to improve the program.

**Questions:**

1. The method’s upfront, binary choice between CoT and synthesis differs from  tool-use agent that dynamically interleave reasoning steps with tool execution. What is the fundamental advantage of this static choice over a dynamic approach, and should the baselines have included comparisons to such agents to better situate the contributions of PIPS?

   - Yao, S., Zhao, J., Yu, D., Du, N., Shafran, I., Narasimhan, K., & Cao, Y. (2023). ReAct: Synergizing Reasoning and Acting in Language Models. *International Conference on Learning Representations*.

   - Paranjape, B., Korbak, F., Hall, C., Santoro, A., Lynch, C., & Hjelm, D. (2023). ART: Automatic multi-step reasoning and tool-use for large language models. *arXiv preprint arXiv:2303.09014*.

2. Could the authors provide a detailed analysis of the computational overhead? Specifically, what is the average increase in end-to-end latency or total token consumption for PIPS when compared directly against the PoT baseline on the algorithmic tasks?

**Ethical Concerns:**

["NO or VERY MINOR ethics concerns only"]

**Final Justification:**

The author's supplementary experiments on multiple models (including open source models) and method overhead dispelled my concerns about the robustness of the method.

**Limitations:**

yes

**Quality:**

3

**Strengths And Weaknesses:**

### Strengths

- **Dynamic Adaptability**： By using a confidence metric to distinguish between tasks, PIPS selects the most appropriate solution path, which helps avoid wasting computational resources on problems not suited for program synthesis.

### Weaknesses

- **Questionable Generality**: The method's general applicability is uncertain, as its performance improvement on certain models, such as GPT-4.1-mini, is marginal.
- **Limited Feedback Mechanism**: The feedback mechanism is limited because it primarily focuses on surface-level structural issues like syntax errors and hardcoded answers. It may not be able to detect or correct deeper logical errors within a program's algorithm.
- **Computational Overhead**: Compared to single-pass methods like PoT, the iterative generation and evaluation process in PIPS results in higher computational costs and increased latency.

---

> ### Author Rebuttal · Authors · 2025-07-31
>
> We thank the reviewer for the thoughtful comments. We address the reviewer’s specific concerns in detail below.
>
> ## Method Generality
>
> Our method results in only a 0.8% absolute increase in harmonic mean accuracy for GPT 4.1-mini, but this is our worst result out of many strong results across other models. We also emphasize that we follow the setting from the BBEH authors of using the harmonic mean accuracy over tasks which is a particularly challenging metric as it downweights easier tasks and significantly upweights the importance of the hard tasks. Even though harmonic mean accuracy improvement is low for GPT 4.1-mini, we see a 11.3% absolute improvement over PoT on Boolean Expressions and a 33.1% absolute improvement over PoT on Dyck Languages.
>
> In addition, we are adding an open-weights model, Qwen3-235B-A22B-Instruct-2507. We report preliminary results over all of the BBEH tasks for this model and will include the complete results in the camera ready version. We observe a 1.3% absolute improvement in harmonic mean accuracy.
>
> | Dataset              | PoT   | CoT   | PIPS  |
> |----------------------|-------|-------|-------|
> | Word sorting         | 0.669 | 0.750 | 0.750 |
> | Dyck languages       | 0.169 | 0.419 | 0.456 |
> | Object counting      | 0.544 | 0.656 | 0.644 |
> | Object properties    | 0.475 | 0.338 | 0.331 |
> | Boardgame qa         | 0.844 | 0.738 | 0.738 |
> | Boolean expressions  | 0.325 | 0.600 | 0.594 |
> | Buggy tables         | 0.400 | 0.169 | 0.231 |
> | Spatial reasoning    | 0.550 | 0.525 | 0.525 |
> | Multistep arithmetic | 0.544 | 0.531 | 0.494 |
> | Geometric shapes     | 0.175 | 0.244 | 0.244 |
> | Temporal sequence    | 0.338 | 0.294 | 0.300 |
> | Causal understanding | 0.450 | 0.606 | 0.606 |
> | Time arithmetic      | 0.762 | 0.569 | 0.738 |
> | Web of lies          | 0.613 | 0.831 | 0.831 |
> | Sarc triples         | 0.181 | 0.287 | 0.287 |
> | Hyperbaton           | 0.256 | 0.287 | 0.287 |
> | Nycc                 | 0.169 | 0.131 | 0.131 |
> | Sportqa              | 0.163 | 0.281 | 0.281 |
> | Linguini             | 0.144 | 0.181 | 0.181 |
> | Movie recommendation | 0.750 | 0.725 | 0.725 |
> | Zebra puzzles        | 0.581 | 0.706 | 0.706 |
> | Harmonic Mean        | 0.326 | 0.366 | 0.379 |
>
>
>
> We believe PIPS has a low effectiveness with GPT 4.1-mini because the model struggles at revising generated code based on the structural feedback we provide, due to its overall weak performance. For o4-mini, a model which performs better overall in terms of the baselines, we see that PIPS improves over PoT performance by 5.7% absolute harmonic mean accuracy. This trend is repeated for Qwen3-235B-A22B-Instruct-2507, which is also more performant in terms of baselines.
>
>
>
> ## Upfront Choice Between CoT and Synthesis
>
> We decided to use an upfront choice between CoT and synthesis since we observed that most of the reasoning problems we were considering were solvable directly by code when the problem was “algorithmic”. For these cases, the problem was that the model could determine that the problem was solvable via program synthesis, but when prompted to generate the code, it failed to produce well-formed code. Since this was the case, even for a system that can interleave code and CoT based reasoning, the problem with code synthesis would remain.
>
> A natural generalization of PIPS is to enable a dynamic problem-solving approach, perhaps where some subproblems are solved by synthesis and others by CoT. We believe this is an interesting future direction, but we did not find it characteristic of the BBEH problems we studied.
>
> ## Agentic / Iterative Baselines
>
> We note that the paper included an agentic baseline, the Gemini Code Interpreter, which we describe on line 188. To provide another agentic baseline, we have additionally evaluated the smolagents library implementation of a CodeAct [1] agent and we include the results below for Gemini-2.0-flash.
>
> | Dataset              | CodeAct (gemini-2.0-flash) |
> |----------------------|---------------------|
> | Word sorting         | 0.525               |
> | Dyck languages       | 0.044               |
> | Object counting      | 0.000               |
> | Object properties    | 0.006               |
> | Boardgame qa         | 0.325               |
> | Boolean expressions  | 0.256               |
> | Buggy tables         | 0.025               |
> | Spatial reasoning    | 0.287               |
> | Multistep arithmetic | 0.000               |
> | Geometric shapes     | 0.369               |
> | Temporal sequence    | 0.000               |
> | Disambiguation qa    | 0.469               |
> | Causal understanding | 0.512               |
> | Time arithmetic      | 0.256               |
> | Web of lies          | 0.206               |
> | Sarc triples         | 0.331               |
> | Hyperbaton           | 0.025               |
> | Nycc                 | 0.106               |
> | Sportqa              | 0.212               |
> | Linguini             | 0.125               |
> | Movie recommendation | 0.388               |
> | Shuffled objects     | 0.006               |
> | Zebra puzzles        | 0.031               |
> | Harmonic Mean        | 0.040               |
>
> We find that CodeAct performs poorly due to several reasons. One issue is from multiple choice questions where the agent does not respond with a letter even when we explicitly give instructions for answer formatting. The other big problem can be seen for object counting and shuffled objects (two highly algorithmic tasks) where CodeAct performs very poorly due to approaching the problem by performing each low level computation with the Python interpreter and everything else with CoT rather than solving the high level problem with code. We will include CodeAct results for all models for the camera ready version.
>
> We additionally include results for the Buffer of Thoughts [2] method over BBEH as suggested by another reviewer. This method uses other test problems from the same task to help with determining an effective method for solving other problems of the task. Overall, we find that this is highly ineffective for BBEH problems.
>
> | Dataset              | Buffer of Thoughts (gemini-2.0-flash) |
> |----------------------|--------------------|
> | Word sorting         | 0.463              |
> | Dyck languages       | 0.056              |
> | Object counting      | 0.006              |
> | Object properties    | 0.000              |
> | Boardgame qa         | 0.275              |
> | Boolean expressions  | 0.031              |
> | Buggy tables         | 0.000              |
> | Spatial reasoning    | 0.113              |
> | Multistep arithmetic | 0.000              |
> | Geometric shapes     | 0.006              |
> | Temporal sequence    | 0.000              |
> | Causal understanding | 0.362              |
> | Time arithmetic      | 0.100              |
> | Web of lies          | 0.087              |
> | Sarc triples         | 0.194              |
> | Hyperbaton           | 0.013              |
> | Nycc                 | 0.019              |
> | Sportqa              | 0.125              |
> | Linguini             | 0.006              |
> | Movie recommendation | 0.013              |
> | Zebra puzzles        | 0.069              |
> | Harmonic Mean        | 0.026              |
>
> [1] Wang, Xingyao, et al. "Executable code actions elicit better llm agents, 2024." ICML 2024.
> [2] Buffer of Thoughts: Thought-Augmented Reasoning with Large Language Models. NeurIPS 2024.
>
> ##  Limited Feedback Mechanism
>
> When we studied the failure modes of current models when generating code for solving reasoning problems, we found that the overwhelming majority of problems came from failing to generate code and the model resorting to textual reasoning. Since this was the most common issue prior to our work, our primary objective in this paper was to address this problem with a simple, but general framework. With this issue now addressed, we agree with the reviewer that more sophisticated forms of feedback will be necessary to address deeper problems such as logical errors now that the code is well-formed, and we are excited about future work in this direction.
>
> ## Analysis of Computational Overhead
>
> We include the average input and output tokens and the average cost for all methods below using Gemini 2.0 Flash with costs determined based on current Gemini API costs. We find that the iterative approaches (CodeAct and Buffer of Thoughts) increase cost over 10X compared to PoT or CoT, but PIPS is overall only 3-4X more expensive than CoT or PoT while achieving much greater accuracy.
>
> | Method    | Avg. Input Tokens | Avg. Output Tokens | Cost (USD) |
> |-----------|-------------------|--------------------|------------|
> | PoT       | 1115.96           | 1333.98            | $0.0006    |
> | CoT       | 1099.77           | 1475.87            | $0.0007    |
> | CodeAct   | 80137.92          | 4023.36            | $0.0096    |
> | Buffer of Thoughts | 340927.19 | 123655.35 | $ 0.0835 |
> | PIPS      | 11839.09          | 2805.78            | $0.0023    |

---

> > ### Comment · Reviewer_ki2H · 2025-08-04
> >
> > The explanations regarding cost and model address my confusion, and I will raise my rating.

---

### Official Review · Reviewer_oWeE · 2025-07-03

**Clarity:** 3
**Significance:** 2
**Originality:** 3
**Rating:** 5
**Confidence:** 3

**Summary:**

This paper introduces Per-Instance Program Synthesis (PIPS), a novel method that addresses the challenges of using large language models (LLMs) to generate and execute programs for complex reasoning tasks. The authors identify three key limitations of existing approaches like Program of Thought (PoT): the difficulty of deciding when to use program synthesis versus direct inference, the lack of task specifications to guide program generation, and the challenge of interfacing programs with unstructured input data. PIPS addresses these issues through an instance-level confidence metric that dynamically chooses between program synthesis and Chain of Thought reasoning, an iterative refinement process using structural feedback to improve program quality without requiring explicit test cases, and explicit symbolic extraction that converts unstructured input into structured program inputs before code generation. Experiments across 30 benchmarks including Big Bench Extra Hard tasks, visual reasoning, and mathematical problems using three frontier LLMs (Gemini-2.0-Flash, GPT-4.1-mini, and o4-mini) demonstrate that PIPS achieves up to 8.6% absolute improvement in harmonic mean accuracy over PoT and reduces undesirable program generations by 65.1%, while maintaining comparable performance to Chain of Thought on non-algorithmic tasks.

**Questions:**

- Could the authors try making a more comprehensive literature review to polish the related work section?

**Ethical Concerns:**

["NO or VERY MINOR ethics concerns only"]

**Final Justification:**

The authors' response mostly addresses my concern. I remain positive about this paper and I keep my score to show my support for its acceptance.

**Limitations:**

Yes.

**Paper Formatting Concerns:**

No formatting concern.

**Quality:**

3

**Strengths And Weaknesses:**

## Strengths

+ This paper proposes a simple yet effective approach to enable the flexible reasoning of LLMs, considering both programming and CoT, and deciding case by case
+ The paper addresses three fundamental challenges in per-instance program synthesis that have not been systematically tackled before: the open-domain nature of deciding when to use synthesis vs. direct inference, the lack of task specifications for program search, and the challenge of interfacing programs with unstructured data.
+ The paper demonstrates rigorous experimental methodology by evaluating PIPS across 30 diverse benchmarks, which was specifically designed to test advanced reasoning capabilities where current LLMs struggle. The evaluation spans three frontier LLMs (Gemini-2.0-Flash, GPT-4.1-mini, and o4-mini) and shows consistent improvements, with up to 8.6% absolute improvement in harmonic mean accuracy over PoT and a significant reduction in undesirable program generations by 65.1%.

## Weaknesses

Overall, I appreciate this paper's simple yet effective approach, and I am very positive regarding its acceptance.

One weakness that bothers me a bit is that the paper misses the discussion regarding a few related works -- some of them might not be closely related to be used as baselines, but are worth discussing. For example, ViperGPT [1] implements a prominent framework for visual inference via Python execution, addresses remarkably similar problems by using LLMs to compose vision-and-language models into subroutines and generating Python code for visual reasoning tasks. PropTest [2] has also explored improving visual programming through property-based testing. I would suggest that the author should do a more comprehensive literature review over the related works, and update its related work section.

[1] Dídac Surís, Sachit Menon, Carl Vondrick. ViperGPT: Visual Inference via Python Execution for Reasoning.

[2] Jaywon Koo, Ziyan Yang, Paola Cascante-Bonilla, Baishakhi Ray, Vicente Ordonez. PropTest: Automatic Property Testing for Improved Visual Programming

---

> ### Author Rebuttal · Authors · 2025-07-31
>
> We thank the reviewer for their thoughtful comments and for recognizing the effectiveness and originality of our work. Following the reviewer’s comment about improving the breadth of our related work section, and the pointer to two relevant papers, we have revised the related work and believe it is much improved. We will add the following to the current related work section:
>
> > Approaches which prompt an LLM to produce code to solve a problem have been used in several domains beyond math and text-based reasoning questions. ViperGPT and followup work tackle visual question answering problems [1, 2], Voyager applies to game playing [3], and Code as Policies focusses on the application of robot control [4]. Recently, general systems such as CodeAct [5] and OpenCodeInterpreter [6] have been proposed for solving problems via code generation similar to the previously mentioned solutions for each task.
> >
> > [1] Surís, Dídac, Sachit Menon, and Carl Vondrick. "Vipergpt: Visual inference via python execution for reasoning." ICCV 2023.
> >
> > [2] Koo, Jaywon, et al. "PropTest: Automatic Property Testing for Improved Visual Programming." EMNLP Findings 2024.
> >
> > [3] Wang, Guanzhi, et al. "Voyager: An Open-Ended Embodied Agent with Large Language Models." TMLR 2024.
> >
> > [4] Liang, Jacky, et al. "Code as Policies: Language Model Programs for Embodied Control." ICRA 2023.
> >
> > [5] Wang, Xingyao, et al. "Executable code actions elicit better llm agents." ICML 2024.
> >
> > [6] Zheng, Tianyu, et al. "OpenCodeInterpreter: Integrating Code Generation with Execution and Refinement." ACL Findings 2024.
>
> We then additionally include CodeAct as a baseline and results for Gemini 2.0 Flash are included in the response to reviewer ki2H. We see that CodeAct significantly underperforms even PoT. The full results will be included in the camera ready version.

---

> > ### Comment · Reviewer_oWeE · 2025-08-07
> >
> > Thanks for the response. The response mostly addresses my concern. I remain positive about this paper and I keep my score to show my support to its acceptance.

---

### Note · Authors · 2025-08-16

Dear AC and Reviewers,

We thank you for your thoughtful and constructive feedback. To summarize the main changes we have made to the paper and main points we provided in response to the reviewer’s comments:
- **Related Work:** We have expanded the related work section as suggested by Reviewer oWeE.
- **Additional Baselines:**
    - We added results for PoT with retries, CodeAct, and Buffer of Thoughts for gemini-2.0-flash and will add the results for all models. PoT+retries yields only minimal gains, showing PIPS’ improvements do not stem only from iteration.
    - We clarified that the existing Gemini Code Interpreter baseline was already iterative.
- **Open-Weights Model:** We included results for the open-weight model  Qwen3-235B-A22B-Instruct-2507, which show that PIPS shows consistent improvements over PoT and CoT (per Reviewer JESL).
- **Cost and Efficiency:** We provide detailed token and cost information, demonstrating that PIPS achieves the highest accuracy despite 3x less cost than CodeAct and 35x less cost than Buffer of Thoughts (per Reviewers ki2H and 7yLF).
- **Switching Mechanism:** We report additional zero-shot and leave-one-dataset-out switch settings, both of which confirm robustness of the switching approach (per Reviewer JESL). Importantly, switch accuracy on disagreement cases is 65%, a +13.8% improvement over the best static choice (per Reviewer 7yLF).
- **Feedback Mechanism:** Structural feedback resolves the main issue of trivial code we identified in generated code. We acknowledge that more sophisticated feedback (e.g., detecting logical errors) is an important direction for future work, but was not the focus of this paper (per Reviewer ki2H).
- **Method Generality (per Reviewers ki2H and JESL):** In regard to the 0.8% improvement over PoT using GPT 4.1-mini, we note that the harmonic mean accuracy is a strict metric that makes aggregate improvements challenging. We still see a 11.3% absolute improvement over PoT on Boolean Expressions and a 33.1% absolute improvement over PoT on Dyck Languages. We also note that GPT 4.1-mini is not the best model we used and that we see larger improvements for more capable models, such as Qwen3-235B-A22B-Instruct-2507 and GPT o4-mini-high, which have higher baseline performance.

We thank the reviewers and AC again for their time and thoughtful feedback, which has helped us improve our paper.

Thanks again,

The Authors of 8455

---

### Decision · Program_Chairs · 2025-09-17

**Decision:**

Accept (poster)

**Comment:**

This paper introduces Per-Instance Program Synthesis (PIPS), an approach that enhances the reasoning capabilities of large language models (LLMs) for complex tasks. PIPS addresses the limitations of existing methods, like Program of Thought (PoT), by dynamically deciding whether to use program synthesis or direct inference based on a confidence metric. Additionally, PIPS iteratively refines programs using structural feedback and converts unstructured inputs into structured program inputs through symbolic extraction.

The reviewers recognized the significance of PIPS in improving the performance of LLMs in multi-step reasoning, but they also raised concerns regarding the generalizability of PIPS, its computational overhead, and the limitations of its feedback mechanism. During the rebuttal phase, the authors effectively addressed these concerns through additional experiments and analysis.

Based on the reviewers’ feedback and the authors’ thorough responses, we recommend accepting this submission. And we urge the authors to address the reviewers' remaining concern on the significance of performance gain in their camera ready version.